# Non-canonical metabolic pathways in the malaria parasite detected by isotope-tracing metabolomics

Simon A Cobbold[1] iD, Madel V Tutor[1,†], Philip Frasse[2,†] iD, Emma McHugh[1], Markus Karnthaler[1], Darren J Creek[3], Audrey Odom John[4] iD, Leann Tilley[1] iD, Stuart A Ralph[1] iD & Malcolm J McConville[1,*] iD

## Abstract

**The malaria parasite, *Plasmodium falciparum*, proliferates rapidly in human erythrocytes by actively scavenging multiple carbon sources and essential nutrients from its host cell. However, a global overview of the metabolic capacity of intraerythrocytic stages is missing. Using multiplex $^{13}$C-labelling coupled with untargeted mass spectrometry and unsupervised isotopologue grouping, we have generated a draft metabolome of *P. falciparum* and its host erythrocyte consisting of 911 and 577 metabolites, respectively, corresponding to 41% of metabolites and over 70% of the metabolic reaction predicted from the parasite genome. An additional 89 metabolites and 92 reactions were identified that were not predicted from genomic reconstructions, with the largest group being associated with metabolite damage-repair systems. Validation of the draft metabolome revealed four previously uncharacterised enzymes which impact isoprenoid biosynthesis, lipid homeostasis and mitochondrial metabolism and are necessary for parasite development and proliferation. This study defines the metabolic fate of multiple carbon sources in *P. falciparum*, and highlights the activity of metabolite repair pathways in these rapidly growing parasite stages, opening new avenues for drug discovery.**

**Keywords** haloacid dehalogenase; mass spectrometry; metabolite repair; Plasmodium; SHMT

**Subject Categories** Metabolism; Microbiology, Virology & Host Pathogen Interaction

**Mol Syst Biol. (2021) 17: e10023**

## Introduction

Considerable progress has been made in reducing the incidence of malaria over the last decade, although the decline in malaria cases has stalled in recent years and resistance to frontline antimalarials is on the rise (WHO, 2019). Identifying new antimalarials with novel targets therefore remains a priority, and significant investment has been made in expanding the drug development pipeline with novel classes of antimalarials (Antonova-Koch *et al*, 2018; Hooft van Huijsduijnen & Wells, 2018). Metabolic enzymes and metabolite transporters are direct or indirect targets of most of the existing antimalarials and current lead compounds (Cowell *et al*, 2018; Ross & Fidock, 2019). However, the total number of enzymes/transporters that have been rigorously validated as drug targets remains small. A detailed understanding of the metabolism of the different developmental stages of the malaria parasite, *Plasmodium falciparum*, and the host cells within which they live is therefore necessary for informing the development of new antimalarial therapies.

*Plasmodium falciparum* progresses through a number of different developmental stages during its life cycle in the *Anopheles* mosquito and its human host. Infection in humans is initiated by infective sporozoites that develop asymptomatically in the liver, resulting in the release of thousands of merozoites that initiate repeated cycles of infection and asexual replication in erythrocytes (i.e. red blood cells (RBCs)) that cause the clinical symptoms associated with malaria. The *P. falciparum* intraerythrocytic developmental cycle (IDC) takes approximately 48 h and involves the development of the metabolically-active trophozoite and schizont stages, followed by cell division of individual parasites into 16–32 new merozoites. The massive expansion of parasite biomass during development is fuelled by the uptake and catabolism of glucose, as well as a number of other essential nutrients (e.g. amino acids, purines and vitamins) that are either directly scavenged from the RBC or derived from breakdown of RBC haemoglobin and other proteins (Roth, 1990; Atamna *et al*, 1994; Liu *et al*, 2006; Olszewski *et al*, 2009). Considerable progress has been made in delineating key salvage and metabolic pathways involved in *P. falciparum* asexual development, which has formed the basis for genome-scale models of parasite metabolism (Fatumo *et al*, 2009; Plata *et al*, 2010; Bazzani *et al*, 2012; Tymoshenko *et al*, 2013). Despite these advances, > 40% of the protein-encoding genome remains unannotated and a significant fraction of annotated metabolic genes have yet to be assigned to specific metabolic pathways or reactions. In the phylum Apicomplexa, many genes have also been repurposed to fulfil non-canonical

1   Department of Biochemistry and Molecular Biology, Bio21 Institute of Molecular Science and Biotechnology, University of Melbourne, Parkville, Vic., Australia
2   Department of Medicine, Washington University School of Medicine, St. Louis, MO, USA
3   Monash Institute of Pharmaceutical Sciences, Monash University, Parkville, Vic., Australia
4   The Children's Hospital of Philadelphia, University of Pennsylvania, Philadelphia, PA, USA
    *Corresponding author. Tel: +61 3 83442342; E-mail: malcolmm@unimelb.edu.au
    †These authors contributed equally to this work

functions, impeding genomic reconstructions and a systematic understanding of the total metabolic capacity of the pathogen (Oppenheim *et al*, 2014; Ke *et al*, 2015). Finally, enzyme promiscuity and side reactions can result in the production of unanticipated and novel metabolites that can have important roles in regulating cellular metabolism (Linster *et al*, 2013; Bommer *et al*, 2019; Dumont *et al*, 2019), further complicating predictions of enzyme function based on gene homology.

Defining the observable metabolic capacity of key developmental stages of *P. falciparum* and its host cell is required to verify the accuracy of genomic reconstructions and to identify unexpected metabolic pathways and gene functions. A number of approaches have been used to undertake a global analysis of the metabolic capacity of other organisms. For example, a system-wide reverse genetics approach was used to identify the metabolic function or indirect metabolic impact of each gene within *E. coli* (Fuhrer *et al*, 2017). The emergence of genome-wide disruption libraries in *P. falciparum* and *P. berghei* makes this approach theoretically possible (Bushell *et al*, 2017; Zhang *et al*, 2018). However, these studies have highlighted the essentiality of many metabolic enzymes in *Plasmodium* spp., limiting the effectiveness of this approach. Even viable but slow-growing mutants generated through such approaches are likely to be difficult to compare directly to parental parasites at a metabolic level. The converse approach – acquiring untargeted mass spectrometry data and verifying the "observed" metabolome – has not yet been fully exploited because of the lack of well-established pipelines for data filtering and metabolite identification. Current liquid chromatography–mass spectrometry platforms allow detection of > 10,000 mass-to-charge ($m/z$) features, yet a significant majority (> 90%) of these features correspond to background noise or degeneracy (Creek *et al*, 2011; Mahieu & Patti, 2017; Wang *et al*, 2019). The absence of autonomous methods for controlling the false discovery rate has hampered the compilation of an accurate metabolome for most organisms to date.

Here we use stable-isotope resolved metabolomics to prioritise $m/z$ features corresponding to metabolites actively synthesised by *P. falciparum* or the host RBC (Huang *et al*, 2014; Sevin *et al*, 2017). Previous work has demonstrated the ability of this approach to define the extent of active ${}^{13}$C-glucose metabolism in RBCs (Srivastava *et al*, 2017), and here we expand this approach to ten biologically relevant ${}^{13}$C-substrates in *P. falciparum*-infected RBCs. Filtering for actively-labelled metabolites enabled > 95% of $m/z$ features to be removed, and the remaining $m/z$ features were then identified and the active metabolome defined. This approach led to the identification of 577 metabolites in uninfected human RBCs and 911 metabolites in *P. falciparum*-infected RBCs corresponding to 41% coverage across the predicted metabolome of *P. falciparum* (the summation of all expected metabolites from all known pathways inferred from a genomic reconstruction irrespective of enzyme gaps). The pattern of stable-isotope labelling for each metabolite allowed us to further infer metabolic reactions corresponding to 70.5% coverage across predicted reactions in *P. falciparum*, with the mis-match between metabolite and reaction coverage largely due to a subset of metabolites participating in many reactions. Defining the "observed" metabolome without constraining the results to the expected composition inferred from genomic reconstructions revealed 89 metabolites and 92 reactions not predicted from genomic reconstructions. These studies have highlighted

unanticipated complexity in *P. falciparum* metabolism, including the presence of active metabolite damage and repair systems in rapidly dividing parasite stages.

# Results

## Global stable-isotope labelling filters for metabolites actively synthesised in uninfected and *P. falciparum*-infected human erythrocytes

*Plasmodium falciparum* trophozoite-infected red blood cells (iRBCs) or uninfected RBCs (uRBCs) were metabolically labelled for 5 h in parallel cultures containing different ${}^{13}$C-labelled compounds. Metabolites were extracted and analysed in parallel by GC-MS, LC-MS polar and LC-MS apolar analytical platforms to maximise coverage of different metabolites classes. All mass-to-charge ($m/z$) features were extracted and untargeted isotopologue grouping performed to identify $m/z$ features that correspond to metabolites that were differentially labelled between iRBCs and uRBCs (Fig 1A). Metabolites were provisionally identified based on METLIN database matching and their identities subsequently confirmed based on comparison with authentic standards, MS/MS matching, stable-isotope incorporation pattern and exact mass. The resulting list of metabolites and their respective labelling patterns were compiled into the "observed" metabolome of *P. falciparum* at the trophozoite stage (Dataset EV1).

As an example of the workflow, polar LC-MS analysis of ${}^{13}$C-glucose labelled iRBC extracts revealed that 859 of the original 33,691 $m/z$ features detected in unlabelled iRBCs exhibited decreased intensity following ${}^{13}$C-glucose labelling (Fig 1B), indicating that they likely correspond to mono-isotopic masses (i.e. the unlabelled species) that decrease as metabolites become enriched for ${}^{13}$C atoms. These $m/z$ features were then ranked by fractional enrichment and the exact mass of the unlabelled feature queried against the METLIN metabolite database (Fig 1C). Of the 859 m/z features, 410 returned putative matches within a 10 ppm tolerance for M-H precursors and are annotated as "presumptive" features. Presumptive features exhibited a broad range of fractional enrichments, ranging from 0.01 to 0.999, indicating no significant bias in annotating putatively labelled metabolites via stable-isotope enrichment. Highlighted is the presumptive feature $m/z$ 275.0167 and its isotopologue group (Fig 1D), matching to three possible metabolites (2-carboxyarabinitol 5-phosphate, 2-carboxyarabinitol 1-phosphate and 6-phosphogluconate; all Δ2 ppm). MS/MS spectral matching, together with the predominance of +6 mass isotopomer in ${}^{13}$C-glucose labelled cells, confirmed the identity of this metabolite to be 6-phosphogluconate (Fig 1E). This procedure resulted in the identification of 232 polar metabolites that were significantly labelled with ${}^{13}$C-glucose and was repeated for all labelled substrates and MS platforms.

## The observed metabolome of uninfected and *P. falciparum*-infected human erythrocytes

The consolidated list of all ${}^{13}$C-labelled metabolites detected on the three MS platforms was then incorporated into the observable metabolome for uRBC and iRBC (Fig 2A). To capture additional metabolites that were not labelled with any of the ${}^{13}$C-substrates tested, all $m/z$

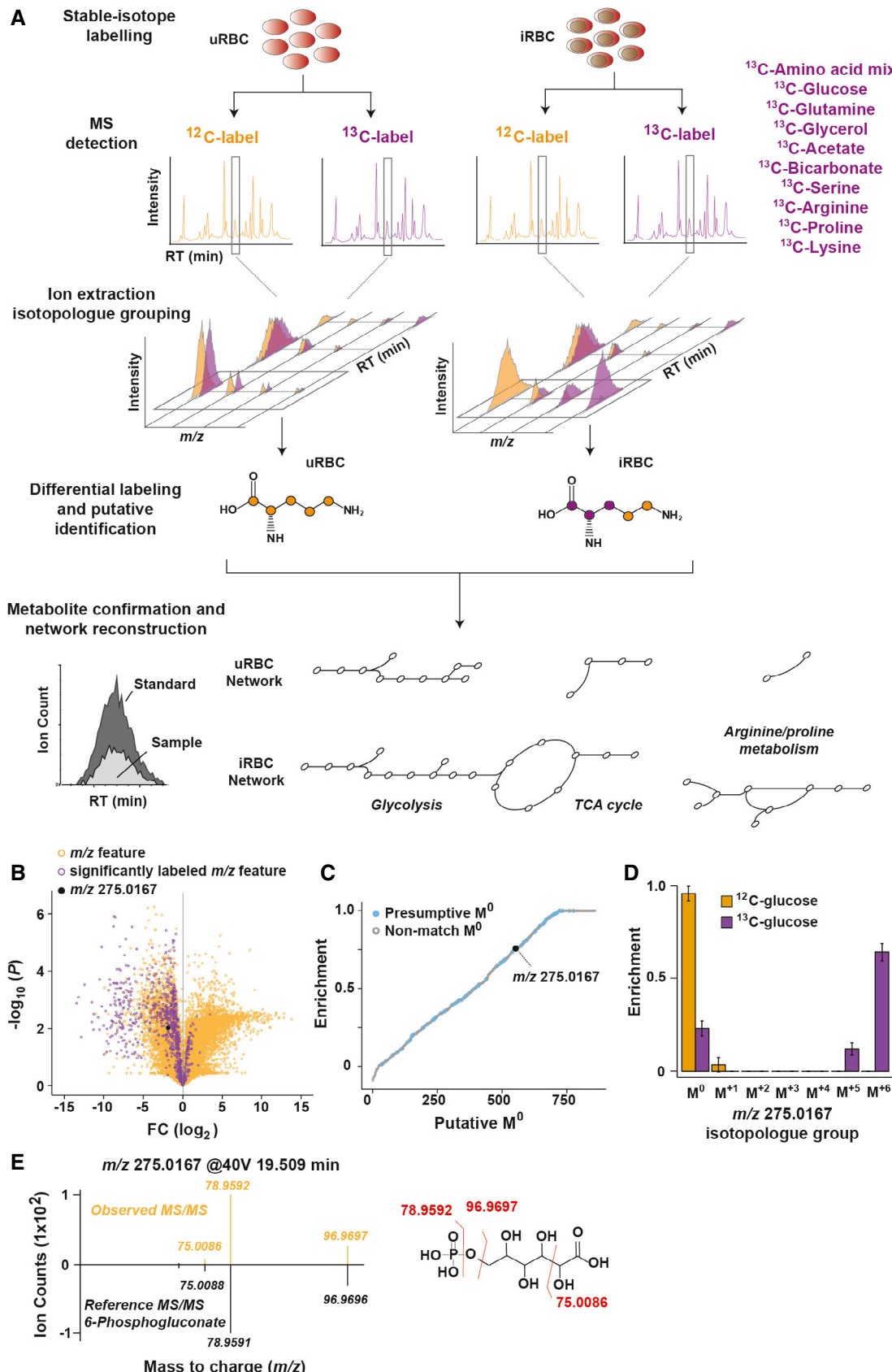

**Figure 1.**

◀

**Figure 1. Strategy for defining the metabolic capacity of *P. falciparum*-infected red blood cells (iRBCs) and uninfected red blood cells (uRBCs) using untargeted stable-isotope labelling.**

A   Purified trophozoite-stage iRBCs and matched uRBCs were labelled with one of ten $^{13}$C-substrates (listed in purple). All mass-to-charge (*m/z*) features identified by LC-MS were extracted and isotopologues grouped. $^{13}$C-labelled and $^{12}$C-labelled isotopologue groups were compared to identify *m/z* features in each cell type and that were differentially labelled between iRBC and uRBC. Putative metabolite identities were then confirmed with authentic standards, MS/MS spectral matching, $^{13}$C-labelling profile and exact mass. The observed metabolic network of each cell type was then constructed using the draft metabolome and $^{13}$C-labelling information and compared to the predicted *P. falciparum* metabolome reported by Huthmacher et al, (2010).

B   All *m/z* features detected from iRBC extracts plotted as the log$_2$ ratio of their abundance detected in $^{13}$C-glucose versus $^{12}$C-glucose conditions. 859 *m/z* features (from a total 33,691 *m/z* features identified) were significantly altered in $^{13}$C versus $^{12}$C samples of iRBC. Highlighted in black is a single *m/z* feature (*m/z* 275.0167).

C   The 859 significant *m/z* features (x-axis; putative M$^0$ species) were ranked by their fractional enrichment (y-axis) and the observed mass queried against the METLIN metabolite database. *m/z* features that returned a hit were classified as presumptive features (blue) and retained for validation. Presumptive *m/z* features displayed a wide range of fractional enrichments, indicating no systematic bias.

D   Changes in the isotopologue group distribution in $^{12}$C-glucose and $^{13}$C-glucose labelled samples of *m/z* 275.0167. Data are presented as the mean $^{13}$C-fractional enrichment ± SEM from six biological replicates.

E   MS/MS fragmentation of *m/z* 275.0167 matched the reference spectrum of 6-phosphogluconate, consistent with the exact mass and labelling pattern, confirming the metabolite identity. This approach was repeated for all presumptive *m/z* features.

features from unlabelled uRBC and iRBC extracts were compared with the expected exact mass of all metabolites in the predicted *P. falciparum* and RBC metabolomes (Huthmacher *et al*, 2010). Putative matches were then verified as described above. The metabolic network reconstruction reported by Huthmacher and colleagues contains predictions of both host and parasite metabolic activity, incorporates literature and manual curation for higher accuracy predictions, and is structured for matching enzyme classification reactions and PlasmoDB IDs. The reconstruction contains 566 metabolites and 349 reactions in human erythrocytes and 1,622 metabolites and 998 reactions in *P. falciparum* (Fig 2A).

Following manual curation and verification, we compiled an "observable" metabolome of iRBCs and uRBCs, which comprised 911 and 577 metabolites, respectively. All metabolites were collapsed into unique KEGG IDs and compared to the predicted metabolome of each cell type (396 and 299 metabolites for iRBC and uRBC, respectively). 255 observed metabolites (with unique KEGG IDs) matched to the predicted metabolome of iRBCs, corresponding to 41% coverage of the predicted metabolome, whereas 152 observed metabolites matched the predicted metabolome of uRBCs (36.3% coverage). Core metabolic pathways were statistically over-represented in the observable metabolomes of both iRBC and uRBC (Fig 2B), with consistently higher metabolite coverage across each pathway for iRBC (Dataset EV2).

Metabolites predicted from genome annotations but not observed in iRBC samples could reflect: (i) incorrect annotation of the genome (e.g. the annotated branched-chain amino acid degradation pathway is likely missing and the sole annotated enzyme in the pathway, BCKDH, is known to fulfil an alternative function; Oppenheim *et al*, 2014), (ii) down-regulation of the metabolic pathway during tropho-zoite development (e.g. *de novo* fatty acid biosynthesis is known to be down-regulated in the presence of exogenous fatty acids in the intraerythrocytic stages; Yu *et al*, 2008) or (iii) technical issues such as low metabolite abundance, sequestration by host/parasite proteins or incompatibility with the applied extraction or MS meth-ods for detection (e.g. haem and ubiquinone biosynthesis). Pathway enrichment analysis of predicted metabolites that were not observed yielded no statistically-enriched pathways (Dataset EV2).

We were interested in defining which metabolites were uniquely detected in iRBCs or uRBCs. iRBCs contained 102 unique KEGG IDs corresponding to 339 metabolites that were not observed in uRBCs,

with phospholipid and CoA biosynthetic pathways most enriched (unadjusted $P = 0.017$ and 0.033, respectively). These pathways are not active in human RBCs, but are required for the rapid growth of *P. falciparum* asexual stages (Fig 2B). Interestingly, five metabolites were only detected in uRBCs. These included N-acetylman-nosamine, sucrose, 5-formyl-tetrahydrofolate, glucosamine and pyri-doxine-5-P. Pyridoxine-5-P is an essential vitamin/cofactor required for the activity of multiple enzymes in both cell types. The absence of detectable pyridoxine-5-P in iRBCs may reflect the sequestration of this cofactor by parasite enzymes (Fig 2C).

A significant number of observed metabolites did not match to the predicted metabolomes of iRBCs and uRBCs (141 and 147, respectively). These unpredicted metabolites did not statistically over-represent any conventional metabolic pathways (Dataset EV2). For example, $^{13}$C-glucose incorporation was observed into the non-canonical glycolytic metabolites, glycero-P-glycerol and acetyl-P in iRBC, along with incorporation into canonical glycolytic and pentose phosphate pathway (PPP) intermediates (Appendix Fig S1). The identification of $^{13}$C-incorporation into unpredicted metabolites highlighted the presence of unanticipated enzyme activities. Strik-ingly, many of the observed metabolites not predicted from genomic reconstructions corresponded to non-canonical metabolites gener-ated by enzyme side reactions or "damaged" metabolites generated by non-enzymatic processes (i.e. oxidation of methionine to methionine sulphoxide; Dataset EV2). Examples of the former include P-lactate and 4-P-erythronate that are formed when enzymes consume their non-preferred substrate (Dumont *et al*, 2019). Glycero-P-glycerol is another non-canonical metabolite that is formed during lipid biosynthesis (Fig 2D) and correlates with high glycolytic flux (Hutschenreuther *et al*, 2013).

We sought to explore in more detail how parasite infection leads to changes in host cell metabolism by comparing the metabolite pool sizes between each cell type (Fig 2E). Metabolites associated with nucleotide biosynthesis, arginine metabolism and phospholipid production were all significantly elevated in iRBCs compared to uRBCs (Dataset EV3), consistent with high rates of synthesis of these metabolites and the need to accumulate biomass during para-site development. However, amino acids and purines are main-tained at comparable levels in both cell types, while intermediates in both glycolysis and PPP intermediates were significantly reduced in iRBC (Appendix Fig S3). Glycolytic and PPP flux is increased up

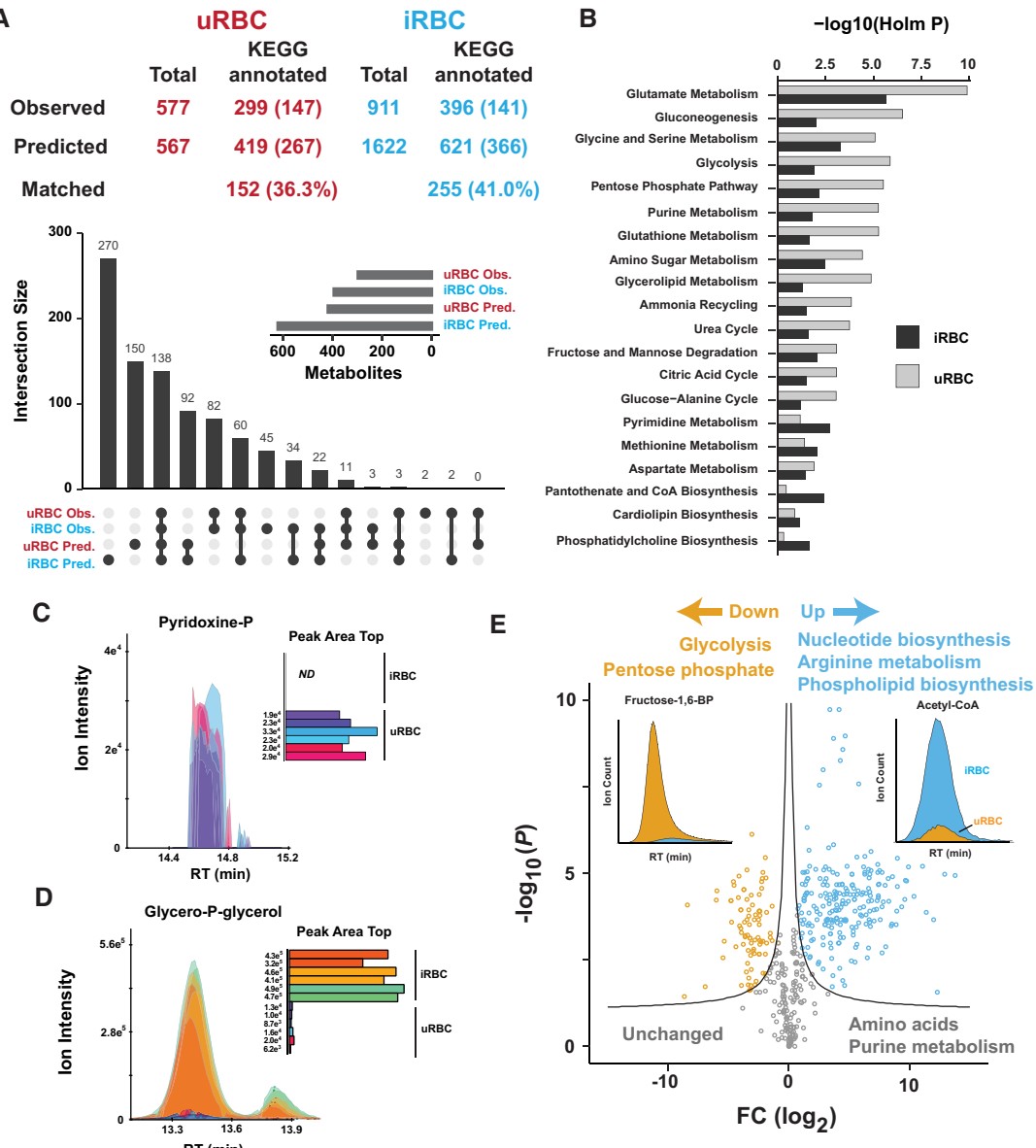

**Figure 2. The observable metabolome of human uninfected RBCs and RBCs infected with *P. falciparum*.**

A The total observed and predicted metabolome of uRBC and iRBC. The total number of metabolites and those with a unique KEGG identification number are reported. Numbers in parentheses refer to unmatched metabolites that were predicted from the genomic reconstruction but not observed or observed metabolites that were not predicted. The total number of metabolites common to both predicted and observed lists (percentage in parentheses) is referred to as "matched". An UpSet plot summarising the overlap of metabolites present in the predicted and observable metabolomes across uRBCs and iRBCs, with the inset depicting the metabolite size of each cell type.

B Metabolic pathway enrichment analysis for the observed metabolome of uRBCs and iRBCs. Unique KEGG ID metabolites were queried against the Small Molecular Pathway Database with $-\log_{10}$ (Holm *P*) reported for the number of metabolite hits reported for each pathway.

C Extracted ion chromatogram of pyridoxine-P, one of five metabolites detected exclusively in uRBCs. X-axis corresponds to retention time (RT), and y-axis corresponds to arbitrary ion intensity of the extracted peak. The inset indicates the integrated peak area for each biological replicate. ND indicates "not detected".

D Extracted ion chromatogram of glycero-P-glycerol, an observed metabolite that was not predicted from genomic reconstructions and is significantly elevated in iRBC compared to uRBC. X-axis corresponds to retention time (RT), and y-axis corresponds to arbitrary ion intensity of the extracted peak. Inset indicates the integrated peak area across each cell type and biological replicate.

E The abundance of all detected *polar* metabolites was compared between iRBCs and uRBCs and presented as the $\log_2$ fold change (iRBC/uRBC) with respect to the $-\log_{10}P$. Significance cut-off was set with $-\log_{10}$ (0.05) + c/(x − x0). Each data point represents a single metabolite, and the insets depict the extracted ion chromatograms for two metabolites that were either elevated (acetyl-CoA) or decreased (fructose-1,6-bisphosphate) in iRBC relative to uRBC.

to 100-fold and 78-fold, respectively, in trophozoite stage-infected RBCs (Roth, 1990; Atamna *et al*, 1994), highlighting the lack of correlation between metabolite abundance and corresponding metabolic fluxes. The reduced pool size of glycolytic/PPP intermediates in iRBC may increase the sensitivity of parasite pathways to subtle changes in exogenous glucose levels by aligning substrate

levels more closely to the respective $k_m$'s of rate-controlling enzymes (Bennett *et al*, 2009; Park *et al*, 2019).

## The metabolic activity network of *P. falciparum* reveals a plethora of metabolic damage-repair systems

To further define the unpredicted metabolic activity of *P. falciparum* and the host RBC, all metabolites labelled with the different [13]C-substrates were mapped to pathways to identify all observable reactions from both iRBC and uRBC. In all cases, the number of metabolites labelled with each tracer was higher in iRBCs compared to uRBCs, as was the complexity of the corresponding sub-networks (Fig 3A). Strikingly, the majority of the detectable metabolome of uRBCs was unlabelled, consistent with the loss of many enzymes and metabolic pathways in mature erythrocytes (Srivastava *et al*, 2017). In contrast, most metabolites detected in iRBC were labelled with one or more [13]C-substrates (Dataset EV1), indicating a high level of redundancy and metabolic complexity in these intracellular parasite stages. Metabolites in iRBC that did not label with any [13]C-substrate tested (183 compounds) mainly consisted of vitamins (e.g. pyridoxine and riboflavin), purines and specific lipid classes (e.g. sphingomyelins) for which the parasite is known to be auxotrophic and dependent on salvage from the host cell or media in the case of *ex vivo* culture.

We then matched labelled substrates to predicted reactions from the genomic reconstruction. Either the substrate and/or the product of 87.4% of metabolic reactions predicted from the reconstructions was labelled with one or more [13]C-tracers in iRBC (Fig 3B). The discordance between metabolite and reaction coverage (42.6% and 87.6% for iRBC, respectively) is partly due to a small minority of metabolites participating in a large number of reactions (Fig 3C). Removing the top ten detectable metabolites (ATP through to NADPH) from the analysis still gave 70.5% coverage across the theoretical reaction landscape of iRBCs. 104 reactions had no substrate/product detection and corresponded to metabolites that were incompatible with MS detection (low abundance or poorly ionised), such as intermediates of haem biosynthesis, superoxide metabolism and dolichol production (Dataset EV2).

We reconstructed the [13]C-label tracing through the predicted metabolic network and incorporated additional reactions to accommodate labelled metabolites not predicted from the genome reconstruction (Appendix Figs S2–S6). These additional steps were based on the closest conversion from known metabolites and consistency of the [13]C/[12]C fraction label observed between putative substrate/product pairs. The [13]C-glutamine reconstruction indicated operation of canonical pathways, such as the TCA cycle, pyrimidine biosynthesis and glutathione metabolism, but also [13]C-labelling into unpredicted metabolites, indicating the presence of unannotated reactions (Fig 3D; highlighted in green). These unexpected reactions did not correspond to complete biosynthetic pathways, but were either additional steps to predicted pathways (e.g. conversion of γ-aminobutyric acid into succinate semialdehyde) or potential metabolic side reactions leading to the production of non-canonical metabolites, such as 2-hydroxyglutarate, that need to be excreted or catabolised to prevent metabolic dysregulation (Lu *et al*, 2012; Intlekofer *et al*, 2017; Dumont *et al*, 2019).

Metabolic side products constituted the largest group of new and unanticipated metabolites detected in the [13]C-network reconstructions.

Of the 92 reactions that were not predicted, 36 corresponding to metabolic side or repair reactions. Using the network reconstructions, we refined the observed metabolites that were unpredicted to 89, corresponding to the highest confidence assignments (consistency with reaction network) (Dataset EV4). These "damaged" metabolites included methionine sulphoxide (which can be a marker of oxidative stress; Mashima *et al*, 2003), unconventional glutathione adducts (e.g. methyl- and succinyl-glutathione) and pipecolate which arises via lysine degradation.

The [13]C-bicarbonate labelling network was generally in agreement with the predicted reactions (Appendix Fig S4) with the exception of labelling into the unconventional metabolite orotidine. Orotidine was also labelled with [13]C-glucose and [13]C-glutamine, consistent with synthesis via a side reaction of pyrimidine biosynthesis. No reaction has been reported for this metabolite, but it is possibly an overflow metabolite, generated via dephosphorylation of orotidine monophosphate. A number of non-phosphorylated intermediates of central carbon metabolism (mannose, galactose) were also labelled in [13]C-glucose-fed iRBC, indicating synthesis and subsequent dephosphorylation of the cognate sugar phosphates (Dataset EV4). These data suggest that unanticipated metabolites can be generated by specific or promiscuous phosphatases. Candidate enzymes include members of the haloacid dehalogenase (HAD) family of enzymes that have been shown to regulate metabolic fluxes *in vivo* (Guggisberg *et al*, 2014; Guggisberg *et al*, 2018; Dumont *et al*, 2019).

## Haloacid phosphatases regulate parasite metabolic flux

We wanted to explore the role of two uncharacterised HAD phosphatases given the important role this protein family plays in regulating metabolic flux and metabolite repair. We targeted PF3D7_1118400 (annotated as HAD4) and PF3D7_0303200 (annotated as a member of the HAD superfamily, hereon referred to as Lipin) for inducible disruption. Both endogenous loci were targeted via Cas9-mediated double-stranded break towards the centre of each gene and the 3'-half replaced with a recodonised sequence flanked by two loxP sites (Wilde *et al*, 2019). The 3'-end was also HA-tagged and a glmS ribozyme introduced in the 3'-UTR. Transfection of this construct into a 3D7 parasite line expressing an integrated copy of the dimerisable cre-recombinase then enabled inducible excision of the gene upon rapamycin addition or transcript degradation when glucosamine was added. Transfections were performed with linearised rescue DNA template and integration confirmed by PCR (Appendix Fig S7). HA-tagged proteins of the expected size were detected and were efficiently depleted following the addition of 100 nM rapamycin and 2.5 mM glucosamine (Fig 4A). A transgenic line with lactate dehydrogenase 1 (LDH1) under inducible disruption was used as a positive control. Disruption of Lipin significantly impaired parasite growth, whereas loss of HAD4 had no effect (Fig 4B). Loss of LDH1 led to specific metabolic changes, with increased abundance of pyruvate (the enzyme's substrate) and reduced abundance of lactate (Fig 4C), demonstrating the validity of the approach for inferring enzyme function.

Loss of Lipin was associated with elevated levels of ceramides and lyso-phosphatidic acid (LPA) lipid species in iRBC (Fig 4D and Dataset EV5). Sphingosine-1-P and sphinganine-1-P were also elevated, suggesting that this enzyme dephosphorylates diverse lipid

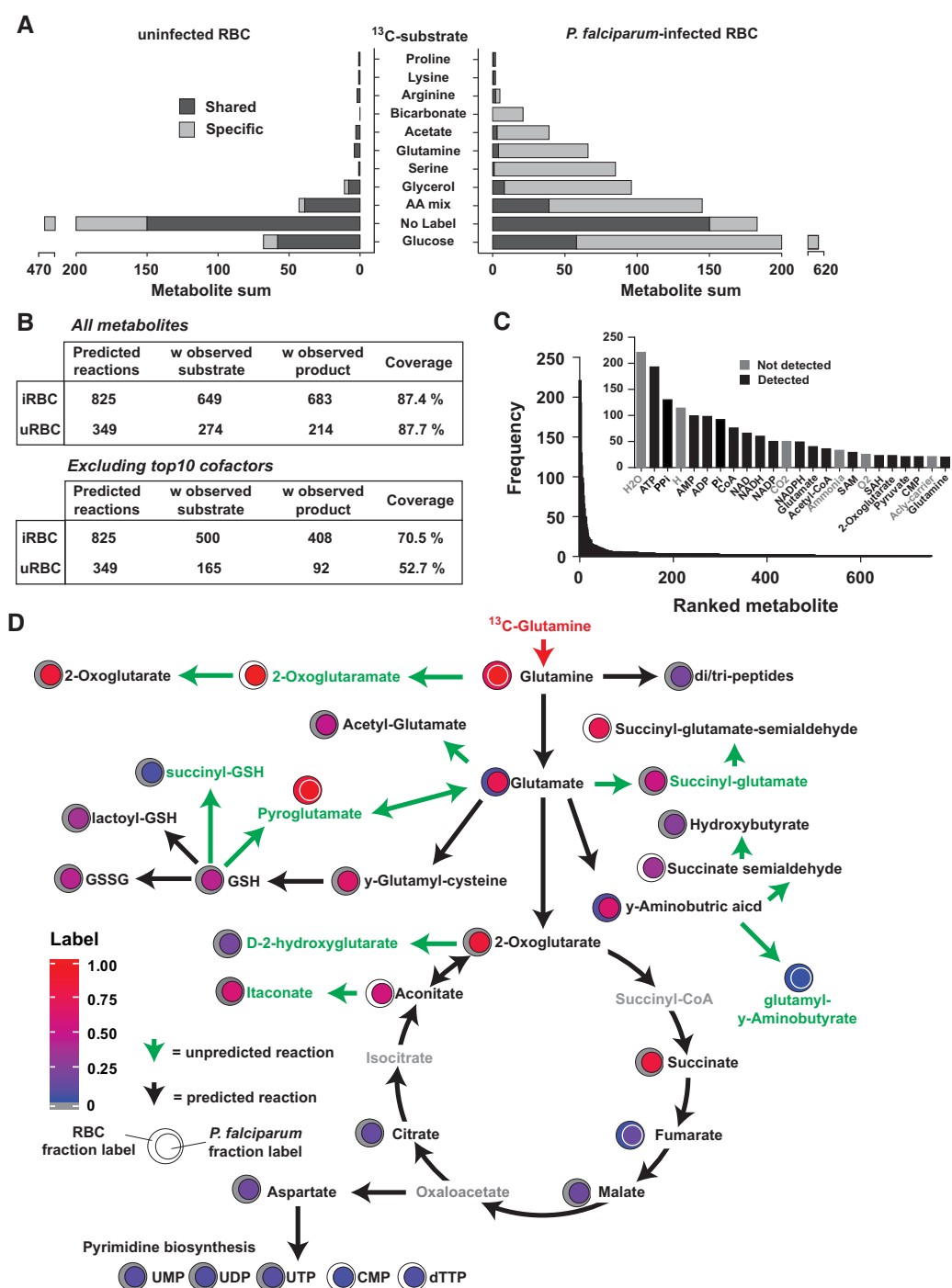

**Figure 3. Reconstruction of the active metabolic networks in *P. falciparum* trophozoite stages.**

A   The sum of metabolites labelled with each ¹³C-substrate in iRBCs and uRBCs. Dark grey indicates metabolites that were ¹³C-labelled in both cell types, whereas light grey represents metabolites ¹³C-labelled in either iRBCs or uRBCs.

B   The number of predicted metabolic reactions in each cell type, the number of reactions with an observed substrate, an observed product, and the number of predicted reactions with no observed substrate or product (unobserved). The same analysis is reproduced following removal of the top ten cofactors present in the observed metabolome.

C   Predicted reactions were deconstructed into individual metabolites and were ranked according to their frequency across all predicted reactions. A small subset of metabolites participate in a large number of reactions. The inset includes ranked metabolites with a frequency (y-axis) ≥ 20 from all predicted reactions. Black indicates metabolites present in the observed metabolome.

D   Reconstruction of the ¹³C-glutamine observed metabolic network. Green metabolites correspond to observed metabolites that are not predicted from the genomic reconstruction. Black arrows indicate where the ¹³C-incorporation is consistent with predicted reactions. Green arrows indicate where the observed labelling pattern is inconsistent with predicted reactions, and proposed reactions are annotated in Dataset EV4.

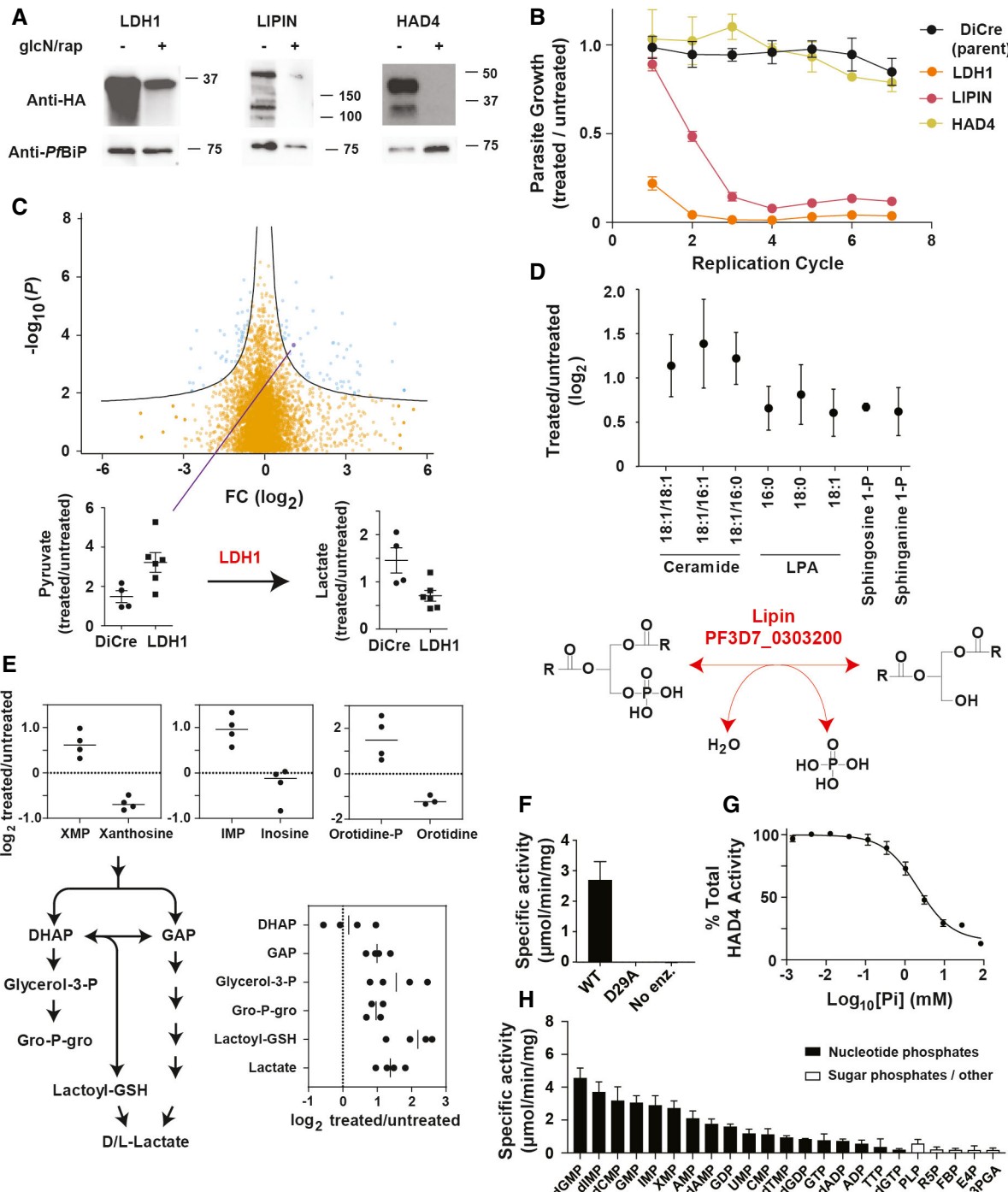

Figure 4.

substrates *in vivo*. This phenotype is consistent with the function of Lipin domain-containing proteins in other eukaryotes, in regulating intracellular pools of phosphatidic acid and diacylglycerols, and flux of these lipids into other pathways of bulk lipid synthesis (phospholipid, triacylglycerol). Interestingly, infection with wild-type *P. falciparum* leads to a depletion of phosphatidic acid and accumulation of diacylglycerol in iRBC (Gulati *et al*, 2015). This likely reflects the need for increased phospholipid biosynthesis in rapidly growing

intraerythrocytic stages, the sequestration of host lipids as triacylglycerols (TAGs) within the parasite, and the pivotal role of phosphatidic acid/DAG signalling in microneme secretion and invasion (Bullen *et al*, 2016). The pleiotropic role of Lipin in these different pathways likely accounts for the severe growth defect observed when Lipin expression was inducibly reduced in this study.

HAD4 disruption led to an accumulation of several nucleotides and glycolytic intermediates which feed into lipid metabolism

**Figure 4. Inducible disruption of haloacid dehalogenase 4 and a putative Lipin.**

A   Anti-HA Western blots for each protein targeted for disruption with a detectable band corresponding to the expected product size. 100 nM rapamycin and 2.5 mM glucosamine were added to induce protein depletion for either one cycle (LDH1) or three cycles (Lipin and HAD4). Anti-*Pf*BiP was used as the loading control.

B   Growth of DiCre-3D7 transgenic lines following LDH1, HAD4 and Lipin depletion with 100 nM rapamycin and 2.5 mM glucosamine was assessed relative to uninduced parasite cultures. The untransfected DiCre-3D7 parasite line was used as a negative control. Data are presented as the mean $\pm$ standard error of the mean (SEM) from three independent replicates.

C   Untargeted LC-MS analysis of LDH1 depletion (one cycle of rapamycin and glucosamine treatment) indicated minimal but selective metabolic changes. *m/z* feature intensities are plotted as the $\log_2$ ratio of treated/untreated and the Benjamini–Hochberg-corrected *P* values across six biological replicates plotted as $-\log_{10}(P)$. Below, LDH1 substrate and product (pyruvate and lactate, respectively) abundance plotted as the ratio of treated/untreated from biological replicates (mean $\pm$ SEM) and the parental DiCre line presented as the negative control.

D   Loss of Lipin (PF3D7_0303200) leads to accumulation of various lipid species (three cycles of rapamycin and glucosamine treatment). Data are presented as the mean $\log_2$ fold change Lipin-depleted (treated) enriched trophozoite-stage iRBCs to untreated controls from three to six biological replicates ($\pm$ SD). The schematic depicts the proposed lipid phosphatase activity of Lipin.

E   Loss of HAD4 leads to increases in intracellular levels of several nucleotides and intermediates in lower glycolysis. Data are represented as the mean $\log_2$ fold change of enriched trophozoite-stage iRBCs treated for three cycles compared to untreated controls from three/four biological replicates. Three nucleotide–nucleoside pairs are depicted in the top inset panels, and HAD4 disruption leads to accumulation of the nucleotide monophosphate and depletion of the nucleoside. The schematic depicts lower glycolysis and triose-phosphate interconversion, with the corresponding metabolite levels following HAD4 disruption depicted in the lower right inset.

F   Phosphatase activity against the generic substrate pNPP of wild-type HAD4, catalytically inactive HAD4$^{D29A}$ and a no-enzyme (No Enz.) control, presented as the mean $\pm$ SEM (three biological replicates).

G   Inhibition of HAD4 phosphatase activity by inorganic phosphate. Phosphatase activity against the generic substrate pNPP as the mean $\pm$ SEM from three independent experiments. The inorganic phosphate IC$_{50}$ of HAD4 is 2.1 $\pm$ 0.21 mM.

H   Substrate specificity of HAD4. Substrates are divided into nucleotide phosphates (black) and other metabolites such as sugar phosphates and vitamins (white). Presented is the enzyme activity from three independent experiments mean $\pm$ SEM). Substrate abbreviations are listed in methods.

(Fig 4E). In order to gain further insights into the potential function of HAD4, we recombinantly expressed and purified HAD4 for analysis of *in vitro* activity (Fig EV1). Phosphatase activity was confirmed (Fig 4F) and inhibited by free phosphate (Fig 4G). HAD4 appeared to dephosphorylate a wide range of nucleotide substrates with a preference for nucleotide monophosphates (Fig 4H). HAD4 may therefore function as a nucleotide monophosphate phosphatase, consistent with accumulation of monophosphates following depletion, including the elevation of orotidine-P and reduced levels of the dephosphorylated orotidine following HAD4 depletion (Fig 4E). The preference of HAD4 towards the monophosphates versus diphosphates (up to sixfold) is similar to the HAD nucleotide monophosphates YrfG present in *E. coli* (Kuznetsova *et al*, 2006). This preference is likely an underestimate as diphosphate nucleotide substrates will produce monophosphate nucleotides in the assay leading to further hydrolysis and higher activity measurements.

Perturbation to nucleotide turnover upon HAD4 loss could indirectly lead to disruption of glycolysis leading to a redirection of flux into lipid precursors. Interestingly, it was previously shown that *P. falciparum* HAD1 also has a direct role in regulating glycolysis (Guggisberg *et al*, 2014), suggesting that this poorly defined family of proteins may have multiple roles in regulating parasite central carbon metabolism.

## AMR1 is necessary for apicoplast function and isoprenoid biosynthesis

AMR1 is an apicoplast-targeted protein which contains a TIM barrel domain with an unknown function and is crucial for apicoplast biogenesis (Apicoplast Minus Rescue 1; PF3D7_1363700) (Tang *et al*, 2019). TIM barrel domain-containing proteins carry out a diverse range of enzymatic reactions and are present in approximately 10% of all enzymes (Goldman *et al*, 2016). The closet sequence similarity is to anthranilate synthase, which is involved in tryptophan biosynthesis. As *Plasmodium* spp. is auxotrophic for tryptophan, we further

investigated the role of AMR1. Inducible disruption of AMR1, with concomitant loss of protein expression (Fig 5A), coincided with impaired proliferation of asexual RBC stages (Fig 5B). AMR1 loss was associated with disruption to isoprenoid biosynthesis and a secondary disruption to haemoglobin catabolism in the digestive vacuole (Fig 5C). This is consistent with a recent study reporting the essentiality of AMR1 for apicoplast biogenesis (Tang *et al*, 2019) and the requirement of IPP for protein prenylation and digestive vacuole formation (Kennedy *et al*, 2019). Curiously, the intermediates formed during the initial steps of the non-mevalonate pathway (DOXP and MEP) are elevated, which is inconsistent with the reduction observed when the apicoplast fails to replicate (Kennedy *et al*, 2019). One interpretation of this phenotype is that AMR1 supplies or regulates the additional cofactors required for these steps. MEP to MEcPP conversion involves three enzymatic steps mediated by IspD, IspE and IspF, which sequentially require CTP consumption, ATP to ADP conversion and CMP release. TIM barrel-containing proteins have a broad function and some members of the family have confirmed roles in nucleotide metabolism (Goldman *et al*, 2016). Recent work demonstrates how the nucleotide pool within the plastid is regulated via pyruvate kinase II (Swift *et al*, 2020) and AMR1 may contribute to maintaining the nucleotide balance necessary for an efficient isoprenoid biosynthetic pathway. Alternatively, AMR1 may have a fundamental role in apicoplast biogenesis and the metabolic phenotype observed is an indirect consequence of impaired apicoplast biogenesis and loss of IPP biosynthetic capacity.

## Mitochondrial serine hydroxymethyltransferase is necessary for normal mitochondrial function

The metabolic network defined by $^{13}$C-serine tracing was the fourth largest sub-network detected in iRBC with 85 metabolites labelled (Fig 6A). The observed $^{13}$C-serine tracing network matched the predicted activity network, including operation of the serine-decarboxylase-phosphoethanolamine methyltransferase pathway which

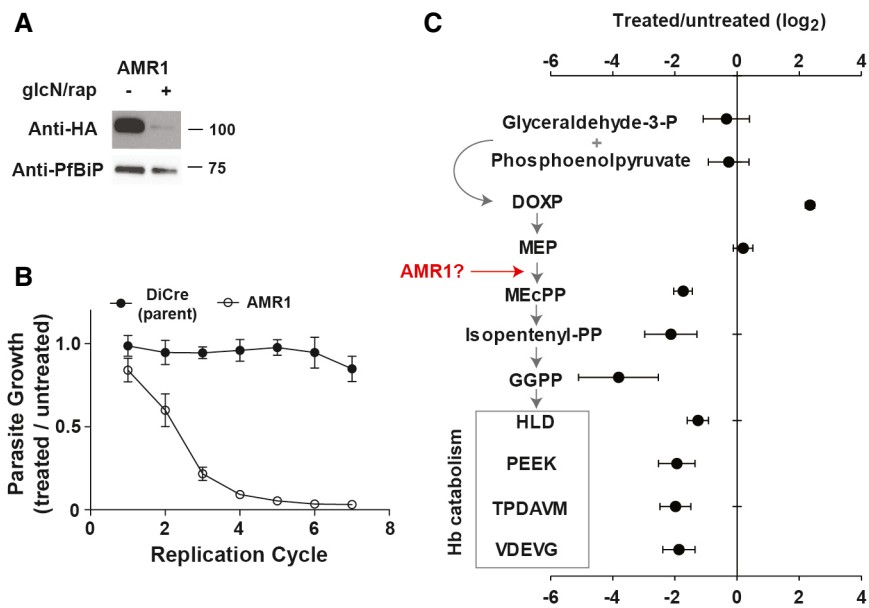

**Figure 5. AMR1 depletion leads to impaired isoprenoid biosynthesis, haemoglobin catabolism and growth.**

A  Anti-HA detection of AMR1 3'-HA tagged in the presence or absence of 100 nM rapamycin and 2.5 mM glucosamine for three cycles. Anti-*Pf*BiP was used as a loading control.

B  Relative parasitaemia was assessed for the untransfected DiCre 3D7 parental line and transfectant parasite lines with AMR1 under inducible disruption. Rapamycin/glucosamine treatment began on cycle zero and parasitaemia determined by flow cytometry and compared to identical lines that were left untreated. Data represent the mean relative parasitaemia ± SEM from three biological replicates (y-axis) across seven replication cycles (x-axis).

C  Metabolite profile following three cycles of rapamycin and glucosamine treatment. The abundance of isoprenoid biosynthetic intermediates (1-deoxy-xylulose-5-P (DOXP), methyl-erythritol-4-P (MEP), methyl-erythritol-cyclo-2,4-PP (MEcPP) and isopentyl-PP) and haemoglobin-like peptides is presented as the mean log₂ fold change from four biological replicates (± SD).

generates precursors for *de novo* biosynthesis of the major phospholipid species (phosphoethanolamine and phosphocholine; Witola *et al*, 2008). However, $^{13}$C-serine labelling suggested that ethanolamine-P may also be directly converted to glycerol-ethanolamine-P rather than being generated via the canonical pathway of PS and PE lipid breakdown (Fig 6A). The identity of the serine phosphate decarboxylase involved in synthesis of ethanolamine-P or the enzyme involved in generating glycerol-ethanolamine-P remains to be discovered (Rontein *et al*, 2003; Liu *et al*, 2018).

The $^{13}$C-serine tracing network also included intermediates in one-carbon folate metabolism. Entry of serine into these pathways is mediated by the enzyme, serine hydroxymethyltransferase (SHMT), which converts serine to glycine with the transfer of one-carbon to tetrahydrofolate. *P. falciparum* encodes two SHMT isoforms: a well-characterised cytosolic isoform (SHMT-C) and a mitochondrial isoform (18% identity to SHMT-C; Maenpuen *et al*, 2009; Pang *et al*, 2009; Read *et al*, 2010). SHMT-C participates in general one-carbon folate metabolism, necessary for dTTP production for DNA synthesis and formylation reactions (Fig 6B). *P. falciparum* lacks the mitochondrial enzymes involved in folate metabolism found in other eukaryotes and no *in vitro* activity could be detected for SHMT-M (Pang *et al*, 2009), leaving the role of the SHMT-M isoform undefined.

To investigate the function of SHMT-M (PF3D7_1456100) and mitochondrial one-carbon metabolism in more detail, we confirmed the localisation of HA-tagged SHMT-M to the mitochondrion

(Fig 6C) (Read *et al*, 2010; Pornthanakasem *et al*, 2012). Conditional disruption of SHMT-M in asexual parasite stages was associated with changes in intracellular levels of many metabolites (Fig 6D and E), in sharp contrast to the limited metabolic changes observed when the essential enzyme LDH1 was depleted (Fig 4C). In particular, loss of SHMT-M was associated with significant changes in intermediates in glycolysis, the pentose phosphate pathway and nucleotide metabolism (Fig 6F; Dataset EV6), suggesting that SHMT-M is fundamental for central carbon metabolism. To differentiate between the possibility that these changes were directly connected to SHMT-M function versus a generalised death phenotype, protein turnover rates were assessed using a recently developed pulse-SILAC method in SHMT-M-depleted parasites (Yang *et al*, 2019). Parasites were labelled with $^{13}$C$^{15}$N-isoleucine for 5 h and the turnover of parasite proteins assessed by measuring the ratio of labelled (newly synthesised) and unlabelled peptides (nascent) in corresponding tryptic peptides (Fig 7A). Similar rates of protein turnover ($R^2 = 0.92$) were observed before or after knock-down of SHMT-M (Fig 7B; Dataset EV7), indicating that parasite protein homeostasis was unperturbed when SHMT-M is disrupted and that the metabolic phenotype is unlikely to be caused by decreased cell viability. Significantly, nuclear-encoded mitochondrial proteins and one of the three mitochondrially encoded proteins (cytochrome b) also exhibited similar rates of turnover before and after SHMT-M knock-down, suggesting that the metabolic phenotype does not reflect marked changes in mitochondrial protein import or translation. This finding indicates

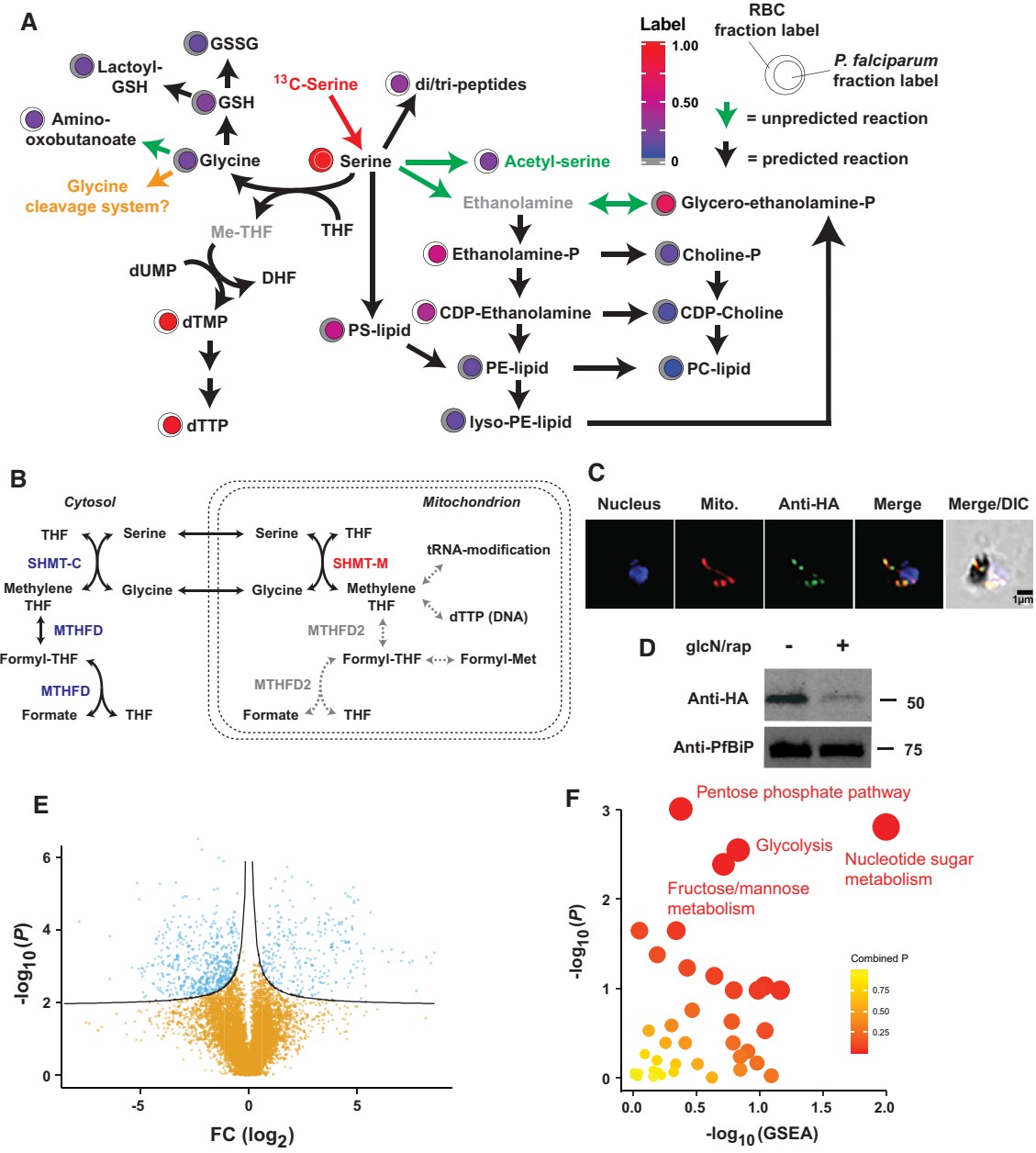

**Figure 6. The architecture and function of the serine metabolic network.**

A $^{13}$C-serine tracing into the metabolic network of iRBC. Black arrows indicate reactions predicted from genomic reconstruction, whereas green arrows indicate reactions that are not predicted but best match the stable-isotope labelling pattern observed. Proposed reactions are annotated below. Metabolites in grey were not observed via mass spectrometry, and green indicates a metabolite observed but not predicted. Highlighted in orange is the glycine cleavage system, which is predicted to be active but has not been verified.

B One-carbon folate metabolism in *P. falciparum*. The parasite encodes all the enzymes necessary for a complete cytosolic pathway but only appears to possess a single enzyme — serine hydroxymethyltransferase — in the mitochondrion. Highlighted in white are the conventional enzymes and reactions of one-carbon metabolism in the mitochondria of other eukaryotes.

C HA-tagged SHMT-M detection via immunofluorescence microscopy. DAPI was used to visualise the nucleus, and MitoTracker CMXros used to illuminate the mitochondrion. Anti-HA signal partially colocalises with the mitochondrion.

D Addition of 100 nM rapamycin and 2.5 mM glucosamine for three growth cycles (+) leads to depletion of the HA-tagged SHMT-M.

E Depletion of SHMT-M leads to significant changes in intracellular levels of many metabolites in iRBC. *m/z* features are plotted as the log$_2$ ratio of treated/untreated intensities and the Benjamini–Hochberg-corrected *P* values across six biological replicates plotted as −log$_{10}$(*P*).

F Pathway enrichment analysis of the *m/z* features identified as significantly different when SHMT-M is depleted. Depicted are the Gene Set Enrichment Analysis (GSEA) *P* values and Mummichog *P* values generated via Metaboanalyst 3.0, with the combined *P* value from each pathway enrichment approach represented by colour.

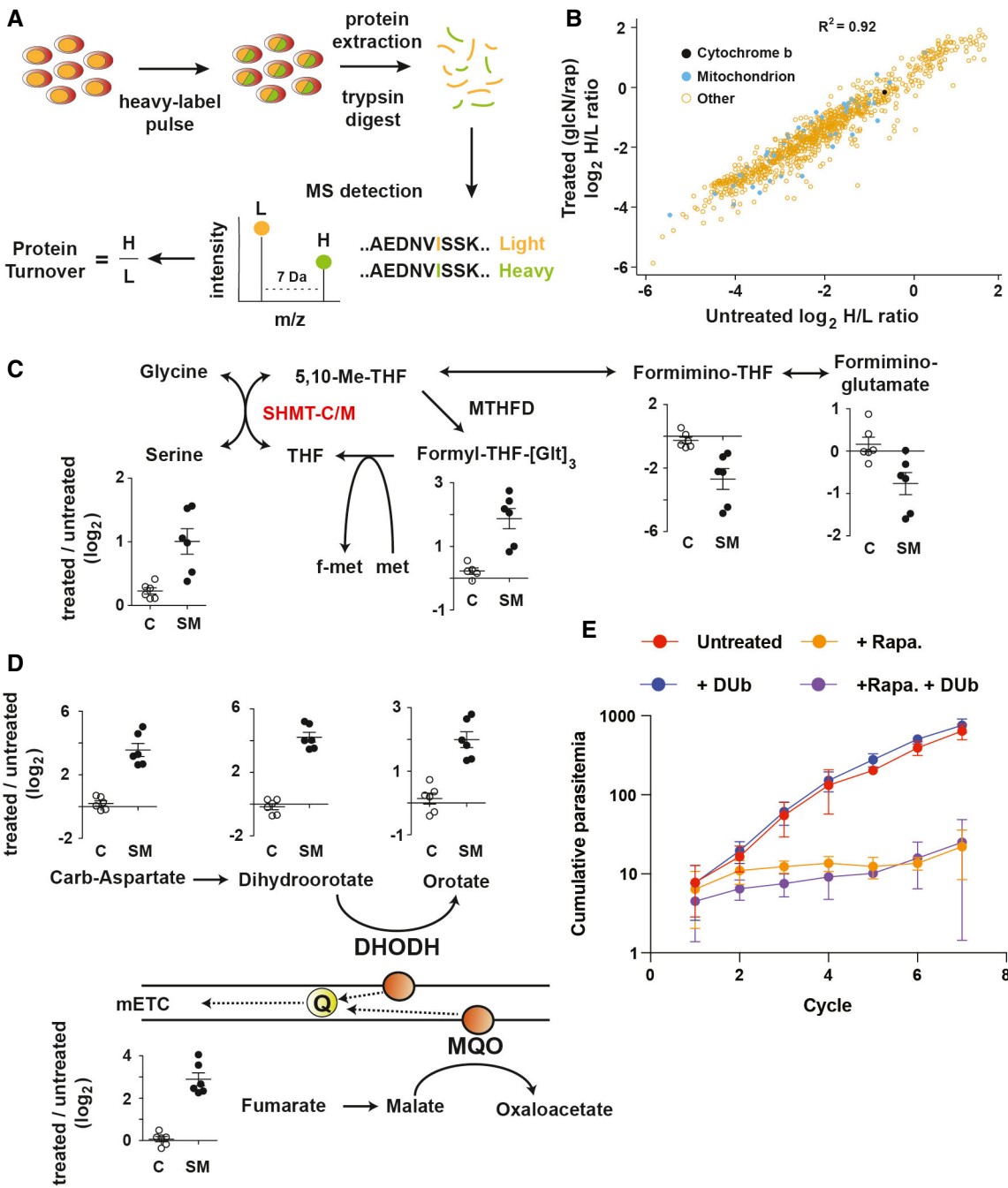

**Figure 7. SHMT-M is required for mitochondrial function, mETC maintenance and asexual growth.**

A *P. falciparum*-infected cultures were incubated for 5 h with $^{13}C^{15}N$-isoleucine (heavy). Following protein extraction and trypsin digestion, each isoleucine-containing peptide exists as a heavy (H) and light (L) form, corresponding to newly translated and nascent pools, respectively.

B The $\log_2$ H/L ratio reflects the turnover (summation of synthesis and degradation) for each protein detected between the control (untreated) and SHMT-M-depleted conditions (thee growth cycles of rapamycin and glucosamine treatment). Data represent the mean $\log_2$ H/L ratio across three biological repeats.

C One-carbon folate intermediates are plotted as the $\log_2$ ratio of their abundance detected in treated (100 nM rapamycin/2.5 mM glucosamine treatment for three growth cycles) compared to untreated in the DiCre parental line (C) and SHMT-M (SM) (mean ± SEM).

D Pyrimidine biosynthetic intermediates accumulate when SHMT-M is disrupted compared to the DiCre parental line (C) following rapamycin/glucosamine treatment (mean ± SEM). Pyrimidine biosynthetic disruption is linked to dihydroorotate dehydrogenase (DHODH) inhibition and blockage of the mitochondrial electron transport chain (mETC). Ubiquinone (Q) is the necessary electron carrier for the mETC.

E Asexual growth when SHMT-M is depleted. Parasitaemia was determined across seven replication cycles (x-axis) via flow cytometry and presented as the mean ± SEM from three independent experiments. Decylubiquinone (Dub) was added at 10 μM to determine if the growth defect observed was due to impaired ubiquinone synthesis or recycling.

                                                        

SHMT-M is not necessary for mitochondrial genome replication (via dTTP synthesis) or tRNA formylation (Fig 6B), in contrast to the function of SHMT2 in mammalian cells (Ducker *et al*, 2016; Minton *et al*, 2018; Morscher *et al*, 2018).

Significantly, knock-down of SHMT-M was associated with changes in one-carbon folate intermediates (Fig 7C), indicating a direct function in one-carbon metabolism. Significant changes were also observed in intracellular levels of pyrimidine biosynthetic intermediates (Fig 7D) which mimicked the metabolic phenotype observed following inhibition of the mitochondrial electron transport chain (mETC) (Ke *et al*, 2015; Cobbold *et al*, 2016). Loss of SHMT-M could lead to loss of ubiquinone – which requires a source of one-carbon units – and is crucial for mETC function. However, attempts to rescue the growth defect of SHMT-M-deficient parasites by supplementation with the ubiquinone analogue, decylubiquinone (Ke *et al*, 2011), were unsuccessful (Fig 7E), suggesting that impaired ubiquinone synthesis is not the primary cause of the growth defect.

We further investigated whether SHMT-M might be in involved in generating one-carbon intermediates for cytoplasmic pathways by labelling iRBC with 2,3,3-D-serine and quantitating the amount of label exported into the cytosol for dTTP synthesis. Negligible amounts of dTTP + 1 isotopomer were detected after 2,3,3-D-serine labelling, indicating minimal export of one-carbon units from the mitochondrion into the cytosol (Appendix Fig S8A), consistent with a lack of a formate transporter at the mitochondrial membrane. SHMT-M is also likely not required for the mitochondrial glycine cleavage system, as $^{13}$C-glycine labelling contributes < 0.1% of the total dTTP pool (Appendix Fig S8B). Taken together, these data suggest that *P. falciparum* SHMT-M may be required to fuel down-stream fluxes, such as the conversion of 5,10-methyl-THF to 10-formyl-THF which can be a major source of NADPH production in eukaryotic cells (Fan *et al*, 2014). A key role in maintaining mitochondrial redox balance would account for the impact of SHMT-M loss on mETC activity, pyrimidine biosynthesis and parasite viability.

# Discussion

We have developed a new approach for measuring *de novo* synthesised and salvaged metabolites in asexual stages of *P. falciparum*, achieving over 70% coverage of all metabolic reactions predicted from *P. falciparum* genome-wide metabolic reconstructions (after exclusion of the top 10 most promiscuous metabolites). We show that asexual parasite stages constitutively salvage a wide range of different carbon sources and nutrients from the host which are assimilated by overlapping anabolic and catabolic pathways, likely contributing to the robustness of parasite metabolism in the RBC. These analyses allowed the detection of over 80 additional reactions and metabolites that are not predicted from gene annotations. Many of these new reactions correspond to additional steps in predicted pathways, or apparent promiscuous-enzyme activity and damage-repair pathways. It is increasingly clear that damaged metabolites have an impact on normal metabolic function – via competitive inhibition or allosteric regulation of key enzymes in central carbon metabolism – and organisms require effective means to detoxify or repair them (Linster *et al*, 2013; Bommer *et al*, 2019; Dumont *et al*, 2019). Exploring the function and impact of the 89 unpredicted metabolites identified in this study will likely reveal an increasingly complex

metabolic arrangement and damage-repair enzymes necessary to maintain parasite metabolism. For example, the $^{13}$C-lysine network included pipecolate which is normally produced during lysine degradation (Appendix Fig S2). *P. falciparum* lacks the genes for a complete lysine degradation pathway but does possess a putative saccharopine dehydrogenase. The subsequent intermediate, aminoadipate semialdehyde, spontaneously cyclises to piperideine-6-carboxylate which competes for pyrroline-5-carboxylate reductase (Fujii *et al*, 2002; Struys & Jakobs, 2010; Linster *et al*, 2013). A pyrroline-5-carboxylate reductase is encoded in the genome of *P. falciparum* and we observed conversion of $^{13}$C-arginine into $^{13}$C-proline consistent with its activity. Curiously, the pyrroline-5-carboxylate reductase itself appears to be essential in *P. falciparum* (Zhang *et al*, 2018), but essentiality of arginine/proline interconversion seems questionable given the excess of each amino acid liberated during haemoglobin catabolism. It is plausible that detoxification of piperideine-6-carboxylate is a necessary function for maintaining parasite metabolism and a primary function of the enzyme (Linster *et al*, 2013).

This dataset comprehensively defines the observable metabolome of uninfected and *P. falciparum*-infected erythrocytes and by using multiple isotopic substrates, enabling the activity of over 70% of predicted reactions to be monitored. We selected the trophozoite stage of asexual development, primarily to capture the greatest number of active reactions at a single stage of development. Investigating other asexual stages may yield additional reactions as well as the process of host metabolome reorganisation. Here we identified five unique metabolites present in uninfected human erythrocytes. Whether these metabolites are actively depleted by the parasite in iRBC remains unclear. Moreover, this work provides a baseline for understanding how environmental factors alter parasite metabolism. For example, phosphate levels in human plasma vary during the day (Lederer, 2014), fluctuate with diet (Goretti Penido & Alon, 2012), and disease severity (Lewis, 1987; Davis *et al*, 1991; Suen *et al*, 2020). Future studies investigating how parasite metabolism responds to fluctuations in plasma phosphate will be of great interest. In addition, the growing evidence that lipid metabolism influences gametocyte commitment necessitates a deeper understanding of how plasma lipids fluctuate during disease progression and how the parasite responds to these triggers and initiates gametocyte commitment (Brancucci *et al*, 2017).

Our attempt to validate the draft metabolome – and the associated reaction network captured via stable-isotope labelling – led us to target several uncharacterised enzymes for inducible disruption. Each of the disrupted genes/proteins contained predicted enzymatic domains, are not part of complete pathways, or are predicted to be essential from genome-wide disruption studies (Bushell *et al*, 2017; Zhang *et al*, 2018). Serine hydroxymethyltransferase is a key enzyme in folate recycling, and the cytosolic version has been well characterised (Maenpuen *et al*, 2009; Pang *et al*, 2009; Read *et al*, 2010). In contrast, the function of the mitochondrion-targeted SHMT has not been defined (Pang *et al*, 2009), and *Plasmodium* spp. lack many of the accessory enzymes needed for mitochondrial folate metabolism. In other eukaryotes, SHMT-M and mitochondrial folate metabolism is important for providing methyl-groups for tRNA modification and effective mitochondrial protein translation and also for providing the precursors for mitochondrial DNA replication (Anderson *et al*, 2011; Morscher *et al*, 2018). Here we demonstrate that mitochondrial protein turnover is unaffected when SHMT-M is

depleted, consistent with earlier studies suggesting that mitochondrial tRNAs are modified in the cytosol prior to import to the mitochondrion (Pino *et al*, 2010). We also show that ubiquinone does not require methyl-group donation via SHMT-M. It remains unclear how dNTPs are sourced for mitochondrial DNA replication, or how methyl-groups are generated and maintained within the mitochondrion for ubiquinone biosynthesis. While metabolite profiling of knock-down parasites confirmed that SHMT-M has a direct role in mitochondrial folate metabolism, significant changes in the levels of many other metabolites from unrelated pathways were also detected. These results strongly suggest that *Plasmodium* SHMT-M has gained additional functions that are necessary for mETC activity and mitochondrial function.

We also performed a preliminary characterisation on two putative haloacid dehalogenases. Lipin was shown to dephosphorylate a subset of lipid species and to be necessary for normal parasite growth. The *P. berghei* ortholog is also predicted to be essential (Bushell *et al*, 2017) but no data are available at other lifecycle stages. It will be of interest to determine the precise lipid substrate specificity of this phosphatase and determine whether the enzyme is necessary for bulk lipid regulation or is necessary for specialise lipid signalling (Gulati *et al*, 2015; Bullen *et al*, 2016). In contrast, HAD4 appears to dephosphorylate a range of nucleotides, with a preference for nucleotide monophosphates. Despite the lack of a growth defect in asexual RBC stages and the dispensability of this protein in *P. berghei* liver stage development (Stanway *et al*, 2019), HAD4 may contribute to optimal metabolic flux and nucleotide homeostasis by repressing monophosphate accumulation. In particular, loss of HAD4 led to an accumulation of orotidine phosphate and a reciprocal decrease in the non-canonical orotidine (Fig 4E), suggesting HAD4 may regulate pyrimidine biosynthetic flux via dephosphorylation of a key intermediate. Lastly, AMR1 appears to be necessary for isoprenoid biosynthesis in agreement with previous work demonstrating its importance to apicoplast biogenesis (Tang *et al*, 2019). Defining the temporal profile of apicoplast impairment will aid in identifying the precise role of AMR1 and the function of other essential proteins involved in apicoplast biogenesis. Collectively, these data show that comprehensive metabolite profiling coupled with multiplex $^{13}$C-labelling can be used to detect subtle, as well as major perturbations in different metabolic mutants and is a powerful tool for functionally defining the large number of poorly annotated genes in these protists.

# Materials and Methods

## Cultivation and stable-isotope labelling

*Plasmodium falciparum*-infected RBCs were cultivated in RPMI 1640 GlutaMAX supplemented with 500 μM hypoxanthine, 22 μg/ml gentamicin, 0.2% (w/v) D-glucose, 25 mM HEPES and albumax II (0.5% w/v). Cultures were routinely synchronised with 5% sorbitol (w/v) to maintain a 12-hour developmental range. Transfectant parasite cultures were maintained with 5% human serum and 0.25% (w/v) albumax II. Uninfected RBCs from the same donor were maintained under identical conditions for 48 h prior to experimentation. Trophozoite-stage *P. falciparum*-infected RBCs were magnetically enriched to > 95% parasitaemia (Colebrook Bioscience) and cell density determined using a Neubauer Haemocytometer. Following a one-hour recovery in fresh RPMI 1640 at 37°C, $1 \times 10^8$

cells were centrifuged and the media replaced with 5 ml of RPMI containing unlabelled substrates or 11 mM $^{13}$C$_6$-glucose, 2 mM $^{13}$C$_5$ glutamine, 1.1 mM $^{13}$C$_6$ arginine, 1 mM $^{13}$C$_3$ serine, 1 mM $^{13}$C$_5$ proline, 5 mM $^{13}$C$_2$ acetate, 23 mM $^{13}$C$_1$ bicarbonate, 1 mM $^{13}$C$_6$ lysine, 10 mM $^{13}$C$_3$ glycerol or $^{13}$C-amino acid mix at 1 mg/ml (and a $^{12}$C-amino acid mix 1mg/ml used as a control). iRBC and matched uRBC samples were incubated for 5 h under standard culturing conditions.

## Metabolite extraction

An aliquot of culture ($1 \times 10^8$ cells/sample) was transferred to a microcentrifuge tube and centrifuged at 14,000 *g* for 30 s. The medium was aspirated, and the cell pellet was resuspended in ice-cold PBS (1 ml) to quench cell metabolism and transferred equally into two microcentrifuge tubes. Following centrifugation (14,000 *g*, 30 s), the PBS was aspirated and 200 μl of 80% acetonitrile (containing 1 μM $^{13}$C$_5$$^{15}$N$_1$ aspartate or 5 μM $^{13}$C$_5$ valine as the internal standard) was added and rapidly mixed for polar LC-MS analysis. Samples were centrifuged to remove precipitated protein (14,000 *g*, 5 min) and the supernatant transferred to a glass mass spectrometry vial (containing an insert) and stored at −80°C until LC-MS analysis. For GC-MS and LC-MS lipid analysis, the quenched cell pellet was suspended in 100 μl chloroform and vortex-mixed, prior to addition of 400 μl of methanol:H$_2$O (3:1 v/v) with further vigorous mixing. Samples were centrifuged for 5 min at 14,000 *g* and the supernatant transferred to a fresh microcentrifuge tube. H$_2$O (200 μl) was added to generate a biphasic mixture which was vortex-mixed and then centrifuged for 1 min at 14,000 *g*, and the top aqueous layer was collected for derivatisation and GC-MS analysis. The lower organic phase was collected for lipid analysis.

For GC-MS analysis, samples were dried by vacuum centrifugation, then resuspended in 90% methanol (100 μl) and transferred to a glass mass spectrometry insert. Samples were dried by vacuum centrifugation, washed with 100 μl methanol and dried. Samples were resuspended in 20 μl of methoxyamine (20 mg/ml) prepared in pyridine, sealed and incubated overnight with shaking at ambient temperature. The next day, 20 μl of BSTFA was added to each sample and incubated for 1 h prior to GC-MS analysis.

Lipid analysis was carried out on the remaining lower organic phase. The organic layer was transferred to a fresh microfuge tube and dried down under nitrogen flow. Dried samples were stored at −80°C until ready for lipid LC-MS analysis.

## LC-MS acquisition

Polar metabolite detection was performed on an Agilent 6550 Q-TOF mass spectrometer operating in negative mode. Metabolites were separated on a SeQuant ZIC-pHILIC column (5 μM, 150 × 4.6 mm, Millipore) using a binary gradient with a 1200 series HPLC system across a 45-min method using 20 mM ammonium carbonate (pH 9) and acetonitrile as outlined in Cobbold *et al*, (2016). Two independent replicates of the metabolite profiling following AMR1 and Lipin depletion were performed using the same ZIC-pHILIC chromatography on a Thermo Q-Exactive operating in both positive and negative mode (rapid switching) as described previously (Creek *et al*, 2016). Lipid extracts were analysed on an Agilent 6550 Q-TOF using the reverse phase chromatography outlined by Bird *et al* (2011).

GC-MS analysis was performed using methods previously described (Saunders *et al*, 2011). Metabolites were separated using a BD5 capillary column (J&W Scientific, 30 m × 250 μM × 0.25 μM) on a Hewlett Packard 6890 system (5973 EI-quadrupole MS detector). The oven temperature gradient was 70 °C (1 min); 70°C to 295°C at 12.5°C/min, 295°C to 320°C at 25°C/min; 320°C for 2 min. MS data were acquired using scan mode with a *m/z* range of 50–550, threshold 150 and scan rate of 2.91 scans/second. GC retention time and mass spectra were compared with authentic standards analysed in the same batch for metabolite identification.

## Draft metabolome compilation

Raw Agilent.d files were converted to mzXML with MSconvert and analysed using the $X^{13}$CMS R package (Huang *et al*, 2014). XCMS centWave peak detection was performed with a 10 ppm mass tolerance, following obiwarp retention time correction. The getIsoLabel-Report function was performed with $P < 0.01$ (Welch's *t*-test), to identify features that significantly labelled under each $^{13}$C-labelling condition in both iRBC and uRBC cell types (isotope mass tolerance = 15 ppm). The getIsoDiffReport function then compared the significantly labelled features between iRBC and uRBC using $P < 0.01$ (Welch's *t*-test). IsoDiffReport *m/z* features were removed if > 1,200 *m/z*, and the putative $M_0$ species queried to the METLIN database with a 10 ppm mass tolerance, excluding all toxicants and adducts, except M-H, from the search. Only *m/z* features with a putative METLIN match were retained for further analysis. *m/z* features were further curated to remove mis-annotated in-source fragments and isotopologues, and where $^{13}$C-enrichment was < 1%. *m/z* features with a retention time of < 250 s were removed from the polar metabolite analysis (assuming these lipid-like species were also detected in the lipid C18 LC-MS analysis). This processing was performed for all $^{13}$C-labelling conditions across both C18 lipid and pHILIC polar LC-MS data and a single metabolite list complied.

Metabolite identification was performed either with retention time matching to authentic standards (approximately 150 in-house metabolite standards or the HMDB compound library), or MS/MS matching. Polar MS/MS data were collected with an auto preferred MS/MS method at a collision energy of 10V, 20V and 40V at a threshold of 10,000 on an Agilent 6545 Q-TOF using the same chromatography outlined above. MS/MS spectra were searched against the METLIN PCDL database via the Agilent Qualitative software package and manually curated. MS/MS spectra without a match were reacquired in positive and negative mode and manually searched against the online METLIN database. Lipid MS/MS data were collected as above and converted to (ABFconverter) and analysed with the MSDial software package (Tsugawa *et al*, 2015).

Polar GC-MS data were analysed using the DeXsi software package (Dagley & McConville, 2018) and metabolite identifications confirmed using an in-house metabolite standards library and merged with the LC-MS datasets.

A first iteration of the observable metabolome of *P. falciparum* was compiled, and unique KEGG IDs (acquired from MetaboAnalyst 3.0 (Xia & Wishart, 2016)) were used to compare to the predicted metabolome of *P. falciparum* reported by Huthmacher *et al* (2010). Under the conditions tested, we could not confidently assign stereoisomers or precise structures (e.g. D-glucose) and assigned KEGG IDs to be consistent with the level of evidence present in the

literature and with the format used by Huthmacher *et al* (2010). uRBC and iRBC datasets were then re-extracted for expected *m/z* of metabolites predicted from Huthmacher *et al* (2010), to assess the rate of false negatives but also because certain classes of metabolites were not expected to label under any of the conditions tested (e.g. vitamins and some cofactors). Putative matches were then verified using authentic standards and MS/MS matching. This iteration of the observable metabolome of iRBC and uRBC cell types was then grouped according to the level of identification: (i) match to authentic standard and MS/MS match, (ii) match to authentic standard or MS/MS match, (iii) $^{13}$C-labelling data consistent with known metabolic pathway and (iv) exact mass match ($\leq 10$ ppm). To capture additional metabolites that were unlabelled across all $^{13}$C-substrates, all *m/z* features from unlabelled uRBC and iRBC extracts were compared with the expected exact mass of all metabolites in the predicted *P. falciparum* and RBC metabolomes. Putative hits were confirmed via pure standards or MS/MS matching where possible.

## Pathway reconstruction

Metabolite enrichment analysis was performed with Metaboanalyst 3.0, and pathway reconstruction was performed for each stable-isotope condition. Observed metabolites that were labelled under a given condition were compared to the theoretical reactions for both iRBC and uRBC (Huthmacher *et al*, 2010). A matched reaction was defined as having a labelled metabolite corresponding to either an expected product or substrate. Unpredicted reactions were reported when an identified metabolite was labelled under a given $^{13}$C-substrate but did not match to a predicted reaction. A putative reaction was proposed based on the nearest proximity to an identified metabolite with consistency in the labelling pattern.

## Generation of transgenic parasites for inducible protein disruption

For each gene of interest, a PAM site was selected at the centre or towards the start of the gene which ranked the highest via CHOP-CHOP v2 (Labun *et al*, 2016). Guide oligos were synthesised for cloning into the BtgZI site of pUF-Cas9g using the In-Fusion cloning kit (Takara). The gene was modified using a rescue template containing a 5'-homology arm immediately upstream of the PAM site, an artificial intron containing a loxP site, a recodonised version of the down-stream gene sequence, a 3xHA tag, which was synthesised by GeneArt. The rescue template was cloned into the pGlmS plasmid using the BglII and SpeI (Prommana *et al*, 2013). The 3'-homology arm was generated via PCR of *P. falciparum* genomic DNA and cloned into the pGlmS plasmid with EcoRI and KasI. Following verification of the correct DNA sequence, both pUF-Cas9g and pGlmS plasmids were purified with a midi-prep kit (Macherey-Nagel). pGlmS plasmid was linearised overnight using BglI, BglII and PvuI. 75–150 μg of pUF-Cas9g and linearised pGlmS were transformed into 3D7 ring-stage parasites with dimerisable Cre-recombinase integrated into its genome (Wilde *et al*, 2019). Cultures were maintained on WR99210 and Blasticidin and complete integration was confirmed on recovered parasites via PCR.

Inducible disruption of each targeted protein was performed cultivating transfected parasites under standard conditions in the presence or absence of 100 nM rapamycin and 2.5 mM

glucosamine. Initial tests were performed to determine length of treatment required to deplete the target protein and effectiveness of the DiCre-loxP and GlmS ribozyme system independently.

## Western blotting

*Plasmodium falciparum* trophozoites were treated with saponin (0.05% w/v) and washed with ice-cold PBS, and centrifuged for 1 min (14,000 $g$) three times. Pellets were then extracted with RIPA buffer and protein concentration determined with a BCA assay. 10 μg of protein per sample was loaded onto a Bis-TRIS Any kDa (Bio-Rad) pre-cast gel and the Precision Plus dual-colour protein ladder as a reference. Protein was transferred onto nitrocellulose membranes using the iBlot2 transfer system and blocked overnight at 4°C with 5% milk powder in TBS-T. Membranes were exposed to an anti-HA antibody (Roche; 1:1,000) overnight at 4°C (in 3% milk powder TBS-T), washed in TBS-T and then incubated with an anti-rat HRP-conjugated secondary antibody (1:5,000; 3% milk in TBS-T) for 1 h. Membranes were then analysed with ECL-reagent (GE Healthcare) with a Bio-Rad imager. Anti-PfBiP was used as the loading control (1:5,000; mouse) and prepared as described above.

## Flow cytometry

Tightly synchronised cultures (final 0.2% Ht and 1–2% Pt) were added to a 96-well plate and incubated at 37°C for 3 h, before washing four times with 200 μl complete RPMI. All conditions were performed as technical duplicates. Parasite viabilities were assessed by each cycle using SYTO 61. For ubiquinone rescue experiments, 10 μM decylubiquinone (Sigma) was added at time zero (when rapamycin and glucosamine were added to half the cultures) and maintained throughout the growth assay.

## Immunofluorescence microscopy

For mitochondrial staining, infected RBCs were stained with 10 nM MitoTracker CMXRos (Invitrogen; M7512) for 30 min at 37°C. Cells immobilised in PHA-E-coated coverslips were then fixed with 2% (v/v) paraformaldehyde/ 0.008% (v/v) glutaraldehyde for 15 min, washed with PBS followed by permeabilisation with 0.1% TX-100 in PBS for 10 min and washed (adapted from (Tonkin *et al*, 2004)). Cells were probed with rat anti-HA (Roche; 3F10) and mouse anti-BiP (Bridgford *et al*, 2018) at 1:1,000 in 3% BSA/PBS. Secondary antibodies used were anti-rat Alexa 647 (Invitrogen; A21247) and anti-mouse Alexa 488 (Invitrogen; A11029) at 1:300 in 3% BSA/PBS. Nuclear staining was performed with 2 μg/ml DAPI and washed prior to mounting. Images were taken using the DeltaVision-Elite (GE Lifesciences) and processed using ImageJ.

## Pulse-SILAC proteomics for quantifying protein turnover

SHMT-M 3D7-DiCre parasites were grown ± rapamycin and glucosamine (100 nM and 2.5 mM, respectively) for three cycles. At trophozoite stage, infected erythrocytes were magnetically enriched and incubated for 5 h in RPMI containing $^{13}C_6^{15}N_1$ isoleucine. Following protein extraction with RIPA buffer (containing protease inhibitors), samples were processed and analysed as outlined previously (Yang *et al*, 2019).

## Recombinant protein expression

*Pf*HAD4 (PlasmoDB ID 3D7_1118400) was amplified from *P. falciparum* genomic DNA using the following primers:

5'-CTCACCACCACCACCACCATATGAAAGATGAACAAATATCATGTTATTATC −3'
5'- ATCCTATCTTACTCACTTATGCAAGTATACTATCTAGATCTCG −3'

The PCR product was cloned by ligation-independent cloning into vector BG1861 (Alexandrov *et al*, 2004), which introduces an N-terminal 6X-His tag. The catalytic mutant HAD4$^{D29A}$ was generated by PCR amplification with the above primers in addition to the following primers:

5'- AATTGATAACGTTCGCCCTTGACCATACGATATG −3'
5'- ATCGTATGGTCAAGGGCGAACGTTATCAATTTTA −3'

These primers were used to introduce an A→C point mutation in the coding sequence. The HAD4$^{D29A}$ construct was cloned by ligation-independent cloning into a modified BG1861 vector that also includes a KFS motif in front of the 6xHis tag to increase protein expression (Verma *et al*, 2019). Constructs were verified by Sanger sequencing.

Constructs were transformed into BL21 (DE3) pLysS *Escherichia coli* (Life Technologies). Cells were grown at 37°C and induced with 1mM isopropyl-β-D-thiogalactoside (IPTG). Cells were harvested by centrifugation and resuspended in lysis buffer containing 10mM Tris–HCL (pH 7.5), 20 mM imidazole, 1 mM MgCl$_2$, 200 mM NaCl, 1 mg/ml lysozyme, 1mM dithiothreitol (DTT) and cOmplete Mini EDTA-free Protease Inhibitor tablets (Roche), and sonicated. Soluble protein was bound to nickel agarose beads (Gold Biotechnology), eluted in 20 mM Tris–HCl pH 7.5, 150 mM NaCl and 300 mM imidazole, and further purified by size exclusion chromatography using a HiLoad 16/600 Superdex 200 pg column (GE Healthcare) equilibrated with 25 mM Tris–HCl (pH 7.5), 250 mM NaCl and 1 mM MgCl$_2$ buffer. Fractions containing HAD4 were pooled, concentrated with a centrifugal filter, flash-frozen and stored at −80°C.

## HAD4 enzymatic assays

All assays were performed in clear 96-well half-area plates using a FLUOstar Omega microplate reader (BMG Labtech) at 37°C. Reaction rates were determined using GraphPad Prism software. All reaction rates represent the mean and standard error of at least three experiments, each with technical replicates. Substrates were purchased from Sigma-Aldrich, except for GTP, dGTP, dATP, dCTP, dTTP, dUTP (Roche), dTMP (BioBasic Canada), DOXP (Echelon Biosciences) and fructose 1-phosphate (Santa Cruz Biotechnology).

Other substrates tested that had activity < 0.1 μmol/min/mg were as follows: cytosine triphosphate; adenosine triphosphate; uridine diphosphate; deoxyuridine triphosphate; deoxycytosine triphosphate; deoxythymidine triphosphate; inosine triphosphate; deoxyadenosine triphosphate; D/L-glyceraldehyde-3-phosphate; sorbitol-6-phosphate; sodium pyrophosphate; deoxyxylulose-5-phosphate; galactose-1-phosphate, sucrose-6-phosphate; 2-deoxy-ribose-5-phosphate; glucose-6-phosphate; glycerol-2-phosphate; mannose-6-phosphate; thiamine monophosphate; fructose-6-phosphate; thiamine pyrophosphate;

sedoheptulose-7-phosphate; racemic glycerol-1-phosphate; 2-phospho-glyceric acid; 2,3-diphosphoglycerate; myo-inositol-2-phosphate; mannose-1-phosphate; trehalose-6-phosphate; glucose-1-phosphate; phosphoenol pyruvate; and dihydroxyacetone phosphate.

Phosphatase activity and phosphate inhibition were measured using the substrate *para*-nitrophenyl phosphate (*p*NPP) (New England Biolabs). Reactions were performed in 50 μl volumes with 1 mM *p*NPP, 50 mM Tris–HCl (pH 7.5), 5 mM $MgCl_2$ and inorganic phosphate (0 mM–42 mM). Reactions contained 1 μg purified recombinant enzyme. *Para*-nitrophenyl production was quantified by absorbance at 405 nm.

Enzyme activity against phosphorylated substrates was measured using the EnzChek Phosphate Assay Kit (Invitrogen) according to supplier instructions. Each 50μl assay contained 200 ng recombinant purified enzyme and 1 mM substrate. The reaction was quantified by absorbance at 360nm, and reactions were linear with respect to time and enzyme concentration.

Enzyme activity against flavin mononucleotide (FMN) was measured using the BIOMOL Green kit (Enzo Life Sciences) to account for the overlap in absorbance between FMN and the EnzChek kit. Each 50 μl assay contained 200 ng recombinant purified enzyme, 25 mM Tris–HCl (pH 7.5), 250 mM NaCl, 1 mM $MgCl_2$ and 1 mM substrate. A time course was taken to obtain kinetic data, and the reaction was quantified by absorbance at 620 nm.

Substrate abbreviations are as follows: dGMP, deoxyguanosine monophosphate; dIMP, deoxyinosine monophosphate; dCMP, deoxycytosine monophosphate; GMP, guanosine monophosphate; IMP, inosine monophosphate; XMP, xanthosine monophosphate; AMP, adenosine monophosphate; 8oxodGMP, 8-oxo-deoxyguanosine monophosphate; dAMP, deoxyadenosine monophosphate; GDP, guanosine diphosphate; UMP, uridine monophosphate; CMP, cytosine monophosphate; dTMP, deoxythymidine monophosphate; dGDP, deoxyguanosine diphosphate; GTP, guanosine triphosphate; dADP, deoxyadenosine diphosphate; ADP, adenosine diphosphate; TTP, thymidine triphosphate; dGTP, deoxyguanosine triphosphate; PLP, pyridoxal-5-phosphate; R5P, ribose-5-phosphate; FBP, fructose-1,6-bisphosphate; E4P, erythrose-4-phosphate; and 3PGA, 3-phosphoglyceric acid.

# Data availability

Mass spectrometry data are available at MetaboLights (https://www.ebi.ac.uk/metabolights/) MTBLS1815 (http://www.ebi.ac.uk/metabolights/MTBLS1815).

**Expanded View** for this article is available online.

## Acknowledgments

We thank Prof. Alan Cowman for plasmids and reagents prior to publication. MJM is a National Health and Medical Research Council Principal Research Fellow. LT is an ARC Laureate Fellow. We acknowledge grant support from the University of Melbourne (SAC), NHMRC (APP1098992 to LT and MJM), the Australian Research Council (DP180102729 to MJM and SAR) and the National Institute of Health (2R01AI103280-06 to AOJ). Bioplatforms Australia provided support of the Metabolomics Australia platform.

## Author contributions

Project design: SAC, SAR, MJM. Mass spectrometry: SAC, DJC. Gene knock-out and protein localisation studies: SAC, MVT. Mutant phenotype analysis: SAC, PF, AOJ, EM, MK. Project supervision and management: SAC, MJM, LT, SAR. Data analysis and manuscript preparation: SAC, MJM. Funding Acquisition: MJM, LT, SAR, AOJ.

## Conflict of interest

The authors declare that they have no conflict of interest.

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
