## [Review Process File · Molecular Systems Biology]

Non-canonical metabolic pathways in the malaria parasite detected by isotope-tracing metabolomics

Simon Cobbold, Madel Tutor, Philip Frasse, Emma McHugh, Markus Karnthaler, Darren Creek, Audrey Odom John, Leann Tilley, Stuart Ralph, and Malcolm McConville

DOI: 10.15252/msb.202010023

Corresponding author(s): Malcolm McConville (malcolmm@unimelb.edu.au), Stuart Ralph (saralph@unimelb.edu.au), Leann Tilley (ltilley@unimelb.edu.au), Simon Cobbold (cobbold.s@wehi.edu.au), Emma McHugh (mchugh.e@unimelb.edu.au), Audrey Odom John (johna3@email.chop.edu), Madel Tutor (mtutor@student.unimelb.edu.au)

Review Timeline:

Submission Date:	24th Sep 20
Editorial Decision:	19th Oct 20
Revision Received:	13th Jan 21
Editorial Decision:	25th Jan 21
Revision Received:	2nd Feb 21
Accepted:	4th Feb 21

Editor: Maria Polychronidou

Transaction Report:

Thank you again for submitting your work to Molecular Systems Biology. We have now heard back from the three referees who agreed to evaluate your study. Overall, the reviewers acknowledge that the study presents a relevant and comprehensive resource for the field. However, they raise a series of concerns, which we would ask you to address in a revision.

I think that the recommendations of the referees are rather clear, and I therefore see no need to repeat any of the points listed below. Please let me know in case you would like to discuss in further detail any of the issues raised. All issues raised by the referees would need to be satisfactorily addressed.

On a more editorial level, we would ask you to address the following points.

REFEREE REPORTS

Reviewer #1:

In the article 'Global analysis of Plasmodium falciparum metabolism using multiplex isotope tracing metabolomics', Cobbold et al., investigate the metabolome of Plasmodium falciparum by analysing infected and uninfected red blood cells (RBCs) using different mass spectrometry platforms and various stable isotope tracers. The use of different platforms enabled coverage of a wide range of metabolites and the use of tracers allowed for the distinction of 'real' metabolites from noise as well as the investigation of specific pathways (e.g. lysine degradation). This thorough analysis has led to the identification of 89 unpredicted metabolites, which are not predicted based on genome annotations and reflect pathways/enzymes, which have not been identified to date or enzymes which exhibit unexpected secondary functions. Cobbold et al. further characterised some of these pathways/enzymes using genetic and molecular biology tools combined with metabolomics and in some cases biochemical characterisation of the enzymes. In doing so, the authors identified the essentiality of Lipin during erythrocytic development and its function in dephosphorylation of several lipids. Furthermore, the authors characterise the role of an apicoplast resident protein,

AMR1, which is potentially involved in isoprenoid metabolism as well as a mitochondrial SHMT, which participates in one-carbon metabolism. Both these proteins were found to be essential and the metabolic phenotype was characterised.

General remarks

This timely study uses state-of-the-art metabolomics and genetic tools. The metabolomics data from the comparison of infected/uninfected RBSs could serve as an important resource for the field. The authors use a thorough untargeted metabolomic analysis to generate hypotheses about unidentified/uncharacterized enzymes and pathways rather than employing metabolomics to characterize and validate the phenotype of a mutant strain. These studies are still rare (one was published earlier this year: Tewari et al., 2020, BMC Malaria Journal) but extremely important to complement genome sequencing studies, genome wide disruption studies, proteomics etc. The approach from Cobbold et al., to utilize a range of stable isotope tracers to identify 'real' metabolites and investigate pathways is elegant and provides more insights than other studies which 'only' determine differences in the abundance of some metabolites in infected/uninfected RBCs. The authors follow up on several findings by targeting enzymes of interest and characterizing the effect on the parasite. This study is of broad interest as it appeals to parasitologist and specifically the malaria community due to its biological significance but also to analytical scientist and the metabolomics community due to the techniques applied here.

Major points:

- In line 242-252 as well as in the subsequent results section, the authors talk about the dephosphorylation of metabolites (e.g. hexoses). Could the authors comment in the text whether these dephosphorylations could also arise non-enzymatically during the metabolite extraction or chromatography or in the ion source and whether the authors can account/correct for this?
- Figure 4A, D-E: Information on how long the parasites were treated with rapamycin cannot be found for the WB and metabolomic analysis. Does the time-point of the WB image in Figure 4A correspond to the time-point of the metabolomic analysis in Figure 4D-E. This should be indicated clearly in the text as well as in the figure legend.
- Figure 4D: While the heatmap is easy to read, could the authors either provide bar graphs which indicate the statistical significance (stdev) or comment on the statistical significance in the main text of the results and/or in the figure legend?
- Line 292-294, the authors speculate that perturbation of nucleotide turnover upon loss of HAD4 could disrupt glycolysis. Do the authors have evidence for this from their metabolomic profiling? Is glycolytic flux impaired although there is no growth phenotype?
- The AMR1 section appears completely detached from the rest of the manuscript. Could the authors include introductory sentences to state how they came about to investigate this protein and how it is related to the metabolome analysis?
- Concerning the Figures associated with AMR1(Figure 5): As for Figure 4, it is not clear for how long the parasites were treated with glcN/rap when the WB image 5A was taken or when the metabolomic data in 5C was acquired. This information should be easily found in the figure legend, figure itself and/or results section as it also crucial for the interpretation of the data.
- Do the authors have data from IFAs to conclude whether the apicoplast is intact at the time of

metabolome analysis of AMR1 deficient parasites? Loss of apicoplast localized protein is typically followed by loss of the organelle which will impact dramatically on the metabolome...

- Figure 5C: As for 4D, could the authors include insights into the statistical significance?
- Figure 6D-F/Figure 7C: can the authors indicate in the figure legend and results section how long parasites were treated for with rapamycin for the WB as well as for the metabolomics data?
- Figure 7C: Does Control 'C' refer to the DiCre parental line treated with rapamycin or to the SHMT-M line untreated? Ideally both controls should have been included.
- In general, where possible, could the authors comment on the role of the investigated enzymes/pathways in other life cycle stages and related organisms? Large datasets are available from genome wide disruption studies in *Plasmodium berghei* and *P. falciparum* (Bushell et al, 2017; Stanway et al, 2019; Zhang et al, 2018) as well as for related organisms (*Toxoplasma gondii*: Sidik et al., 2016). While the authors refer in some cases to these studies, this could be further explored and provide context. E.g. could HAD4, which is dispensable for erythrocytic development be essential in other life cycle stages?

Minor points:

- Check for consistency between British and American English. Authors sometimes refer to 'heme', in other cases to 'haem'
- Line 20: consider changing 'remain' to 'remains'
- Line 104-105: is 'monoisotopic mass' the right expression here? Shouldn't it rather be the 'parental ion, M0'?
- Line 117: consider changing 'label substrates' to 'labelled substrates'
- Line 266: insert space in '100nM'
- Line 268: remove accession number and refer only to 'Lipin' as introduced prior
- Line 269: remove 'loss' after HAD4: 'loss of Had4 loss'

Reviewer #2:

In this manuscript, the authors implement a comprehensive isotope-labeling strategy to broadly profile the small-molecule metabolome actively synthesized during blood-stage infection by *Plasmodium falciparum* malaria parasites. Through use of 10 different ¹³C-labelled precursors, multiple extraction conditions, and data filtering against unlabeled erythrocytes and high-abundance metabolites, the authors provide evidence for novel biosynthetic capability by parasites, some of which they explore in more depth via targeted knockdown of four annotated but poorly studied parasite proteins.

Many prior studies have queried parasite biosynthesis of general or specific metabolites but have generally relied on a single ¹³C-labelled precursor (e.g., glucose) or solvent conditions (e.g., MeOH-soluble metabolites). The present study appears to provide the most comprehensive picture of active parasite-dependent metabolite synthesis in blood-stage trophozoites and will serve as an invaluable resource to the parasitology community, especially if the data is organized and

presented in a community-accessible manner. Modest insights are made with respect to individual pathways, but these advances seem incremental relative to the broad impact this study will have as a comprehensive metabolite resource. Insights provided by the knock-out/down studies of the tested proteins are intriguing but preliminary and not developed in depth.

Major points:

1. As a comprehensive resource, how will the authors make this data available to the community? The searchable human metabolome database (<https://hmdb.ca>) has provided a valuable resource. Can the data from the present study be integrated into PlasmDB.org or elsewhere in an easily searchable format? Such a resource is sorely lacking for the parasitology community.
2. The authors limit their study to trophozoite-stage parasites labelled for 5 hours. How were parasites synchronized, and what is the synchrony window? How does this focus on trophozoites and 5-hour labeling potentially bias or limit the observations regarding active pathways and detectable metabolites? Some discussion of this point in the paper would be helpful to justify the authors' choice of these conditions and to contextualize the present observations into the parasite lifecycle.
3. Related to point #2, the impact of this study as a baseline data-set could be improved and clarified by explicit discussion of frontier conditions or questions that can now be dissected via comparative metabolomics. For example, how does de novo metabolite synthesis versus scavenging change in ring or schizont-stage parasites? Given evidence for metabolite scavenging, how does blocking new permeability pathways (e.g., furosemide) or specific knock-down of Exp2, which was recently suggested (<https://pubmed.ncbi.nlm.nih.gov/30150733/>) to mediate metabolite uptake into parasites independent of its role in protein export, alter metabolic profiling? The authors don't need to perform these studies but some discussion of critical applications of the methods herein to address novel questions moving forward would be helpful and place the impact of the present study and advances in context.
4. The authors' use of dual loxP/Cre recombinase plus glmS ribozyme for gene regulation is interesting. What efficiency for gene excision did the authors observe? Nothing is reported. For the protein knockdown in figure 4A, was gene excision or transcript degradation more critical for the observed loss of protein levels?

Minor points:

5. For figure 4H, the 2-4-fold variation in specific activities observed for HAD4 for the profiled monophosphate versus diphosphate substrates seems negligible and much, much smaller than the orders of magnitude difference in substrate specificities reported for optimized phosphate mono- versus di-esterases (e.g., <https://pubs.acs.org/doi/10.1021/bi060847t>). This small difference raises questions about enzyme purity/activity, the kinetic regime (saturating or sub-saturating) in which substrate activities were profiled, and whether the observed specific activities (as opposed to more rigorous determination of specificity constants, k_{cat}/K_m) are an accurate reflection of substrate specificity. The manuscript does not depend critically on this data but some discussion of these limitations and comparison of observed HAD4 activity to well-characterized enzymes in this superfamily would be helpful.
6. Lines 315-316: A recent study identified the apicoplast-targeted pyruvate kinase II as likely critical for maintaining NTP pools in the apicoplast (<https://pubmed.ncbi.nlm.nih.gov/32815516/>). The authors may wish to mention and cite this study.

Reviewer #3:

The major contribution of this paper is a thorough analysis of the metabolic capacity of Plasmodium-infected red blood cells and providing experimental evidence of a majority of putative metabolic pathways based on gene annotations. This dataset will be an important resource for gene-specific studies of metabolic enzymes, for example as targets of small molecules or identification of new metabolic pathways. Overall this is an important addition to metabolic network analysis of Plasmodium parasites. Several enzymes with unassigned functions are characterized with new hypotheses towards their cellular roles.

Specific comments and questions--

Abstract: The authors state "leading to identification of previously uncharacterized enzymes in isoprenoid biosynthesis, lipid homeostasis, and mitochondrial metabolism." Unfortunately because enzyme depletion can have indirect cellular effects, there is insufficient evidence that the enzyme acts directly in these pathways. Please modify statement as it could be misleading to suggest biochemical activity that has not been proven for these new enzymes.

121-125; Fig2. While the identification of ¹³C-labeled metabolites is well explained, it is not clear how unlabeled compounds are verified. As stated in the introduction, "The absence of autonomous methods for controlling the false-discovery rate, has hampered the compilation of an accurate metabolome for most organisms to date." So absent isotope incorporation how are unlabeled metabolites specifically being identified?

Related to this, how many of the metabolites identified were ¹³C-labeled vs unlabeled?

187-247: Is it possible these side reaction and damaged products are occurring in dying or lysed cells? Please discuss this possibility.

Note: My expertise is in metabolic pathways and Plasmodium parasites] biology. I am less familiar with technical aspects of metabolomics mass spectrometry. Please refer to other experts for evaluating these aspects of the work.

Reviewer #1:

In the article 'Global analysis of Plasmodium falciparum metabolism using multiplex isotope tracing metabolomics', Cobbold et al., investigate the metabolome of Plasmodium falciparum by analysing infected and uninfected red blood cells (RBCs) using different mass spectrometry platforms and various stable isotope tracers. The use of different platforms enabled coverage of a wide range of metabolites and the use of tracers allowed for the distinction of 'real' metabolites from noise as well as the investigation of specific pathways (e.g. lysine degradation). This thorough analysis has led to the identification of 89 unpredicted metabolites, which are not predicted based on genome annotations and reflect pathways/enzymes, which have not been identified to date or enzymes which exhibit unexpected secondary functions. Cobbold et al. further characterised some of these pathways/enzymes using genetic and molecular biology tools combined with metabolomics and in some cases biochemical characterisation of the enzymes. In doing so, the authors identified the essentiality of Lipin during erythrocytic development and its function in dephosphorylation of several lipids. Furthermore, the authors characterise the role of an apicoplast resident protein, AMR1, which is potentially involved in isoprenoid metabolism as well as a mitochondrial SHMT, which participates in one-carbon metabolism. Both these proteins were found to be essential and the metabolic phenotype was characterised.

General remarks

This timely study uses state-of-the-art metabolomics and genetic tools. The metabolomics data from the comparison of infected/uninfected RBCs could serve as an important resource for the field. The authors use a thorough untargeted metabolomic analysis to generate hypotheses about unidentified/uncharacterized enzymes and pathways rather than employing metabolomics to characterize and validate the phenotype of a mutant strain. These studies are still rare (one was published earlier this year: Tewari et al., 2020, BMC Malaria Journal) but extremely important to complement genome sequencing studies, genome wide disruption studies, proteomics etc. The approach from Cobbold et al., to utilize a range of stable isotope tracers to identify 'real' metabolites and investigate pathways is elegant and provides more insights than other studies which 'only' determine differences in the abundance of some metabolites in infected/uninfected RBCs. The authors follow up on several findings by targeting enzymes of interest and characterizing the effect on the parasite. This study is of broad interest as it appeals to parasitologist and specifically the

malaria community due to its biological significance but also to analytical scientist and the metabolomics community due to the techniques applied here.

Major points:

- In line 242-252 as well as in the subsequent results section, the authors talk about the dephosphorylation of metabolites (e.g. hexoses). Could the authors comment in the text whether these dephosphorylations could also arise non-enzymatically during the metabolite extraction or chromatography or in the ion source and whether the authors can account/correct for this?

We agree with reviewer 1 that this is an important point. We were indeed surprised to see multiple labelled neutral sugars in our analyses of ¹³C-glucose labelled parasites, providing direct evidence for the interconversion of Glc6P with other sugar phosphates and their subsequent dephosphorylation. We are confident that these neutral metabolites is not a result of in-source fragmentation of corresponding phosphorylated metabolites as we see no evidence for systematic dephosphorylation of sugar phosphate standards which have been spiked into samples prior to extraction. Moreover, we observed similar levels of free hexoses using both LC and GC platforms with different extraction procedures, sample preparation and most importantly ionisation conditions. Lastly, several of the free hexoses exhibited lower labelling than the phosphorylated species. For example, ¹³C-enrichment in Fru6P was significantly higher than for Fru (85% versus 65%) when parasites were labelled with ¹³C-glucose indicative of a precursor-product relationship (enrichment should be the same if Fru primarily resulted from in-source dephosphorylation of Fru6P). Together these lines of evidence indicate that the free hexoses are endogenous metabolites, almost certainly generated from their cognate sugar phosphates *in vivo* by as yet undefined phosphatases, and not an artifact of sample processing or data acquisition.

- Figure 4A, D-E: Information on how long the parasites were treated with rapamycin cannot be found for the WB and metabolomic analysis. Does the time-point of the WB image in Figure 4A correspond to the time-point of the metabolomic analysis in Figure 4D-E. This should be indicated clearly in the text as well as in the figure legend.

We have amended the text in the figure legend to clarify that the samples for the Western blots were collected at the same time as the metabolomic analysis, which was after three cycles for Lipin- and HAD4- knock-out lines, and one cycle for LDH1 knock-out line.

- Figure 4D: While the heatmap is easy to read, could the authors either provide bar graphs which indicate the statistical significance (stdev) or comment on the statistical significance in the main text of the results and/or in the figure legend?

We have updated Figure 4D to include the standard deviation for each metabolite, and we have replotted 4E to depict each replicate.

- Line 292-294, the authors speculate that perturbation of nucleotide turnover upon loss of HAD4 could disrupt glycolysis. Do the authors have evidence for this from their metabolomic profiling? Is glycolytic flux impaired although there is no growth phenotype?

We show that loss of HAD4 leads to changes in intracellular pools of glycolytic intermediates, indicative of a change in glycolytic flux in mutant lines. However, more detailed quantitative analysis of glycolytic fluxes in mutant lines would require analysis of ¹³C-glucose incorporation into glycolytic intermediates over seconds/minutes, given the very high glycolytic flux in these parasite stages. These analyses are more complex than the steady-state labelling undertaken elsewhere in this study

and we believe, are beyond the scope of this study. However, we would note that we have recently shown that loss of related HAD enzymes does result in complex metabolic changes that include significant changes in glycolytic fluxes, which support HAD4 having a similar role.

- The AMR1 section appears completely detached from the rest of the manuscript. Could the authors include introductory sentences to state how they came about to investigate this protein and how it is related to the metabolome analysis?

We thank reviewer 1 for this feedback. We have changed the text to include a more detailed rationale for why we targeted this gene/protein for disruption.

- Concerning the Figures associated with AMR1(Figure 5): As for Figure 4, it is not clear for how long the parasites were treated with glcN/rap when the WB image 5A was taken or when the metabolomic data in 5C was acquired. This information should be easily found in the figure legend, figure itself and/or results section as it also crucial for the interpretation of the data.

We again thank reviewer 1 for this feedback. We have amended the text in the figure legend to clarify that the western blot presented corresponds to the same stage of knockdown and the metabolomic analysis which was three cycles.

- Do the authors have data from IFAs to conclude whether the apicoplast is intact at the time of metabolome analysis of AMR1 deficient parasites? Loss of apicoplast localized protein is typically followed by loss of the organelle which will impact dramatically on the metabolome...

We agree with reviewer 1 that a broader analysis of apicoplast retention in the AMR1 deficient parasites would be of interest for understanding why loss of AMR1 leads to the perturbation in isoprenoid biosynthesis and loss in viability. As presented in the supplemental dataset, few metabolites are altered when AMR1 is depleted, suggesting this phenotype is unlikely to be caused by generalised cell death. The metabolic changes following AMR1 loss are also distinct from those that we have found to be associated with the delayed death phenotype that occurs when apicoplast biogenesis is disrupted with various anti-malarials (see Kennedy et al. PLoS Biology 2019). Together these data suggest that the perturbation to isoprenoid biosynthesis observed following AMR1 disruption is distinct from the loss of apicoplast/ delayed death phenotype, and is directly related to AMR1 function. We are currently characterizing AMR1 function in more detail, although we believe that it is beyond the scope of this study.

- Figure 5C: As for 4D, could the authors include insights into the statistical significance?

We have updated this figure to include the standard deviation for each metabolite.

- Figure 6D-F/Figure 7C: can the authors indicate in the figure legend and results section how long parasites were treated for with rapamycin for the WB as well as for the metabolomics data?

We have amended the text in the figure legend to clarify that the western blot presented corresponds to the same stage of knockdown and the metabolomic analysis which was three cycles.

- Figure 7C: Does Control 'C' refer to the DiCre parental line treated with rapamycin or to the SHMT-M line untreated? Ideally both controls should have been included.

Yes, clarified in figure legend

- In general, where possible, could the authors comment on the role of the investigated enzymes/pathways in other life cycle stages and related organisms? Large datasets are available from genome wide disruption studies in *Plasmodium berghei* and *P. falciparum* (Bushell et al, 2017; Stanway et al, 2019; Zhang et al, 2018) as well as for related organisms (*Toxoplasma gondii*: Sidik et al., 2016). While the authors refer in some cases to these studies, this could be further explored and provide context. E.g. could HAD4, which is dispensable for erythrocytic development be essential in other life cycle stages?

We have now separated the Results and Discussion sections and included additional discussion on how this study relates to other genome-wide expression and knock-out studies and development of different life cycle stages.

Minor points: Corrected all

- Check for consistency between British and American English. Authors sometimes refer to 'heme', in other cases to 'haem'
- Line 20: consider changing 'remain' to 'remains'
- Line 104-105: is 'monoisotopic mass' the right expression here? Shouldn't it rather be the 'parental ion, M0'?
- Line 117: consider changing 'label substrates' to 'labelled substrates'
- Line 266: insert space in '100nM'
- Line 268: remove accession number and refer only to 'Lipin' as introduced prior
- Line 269: remove 'loss' after HAD4: 'loss of Had4 loss'

Reviewer #2:

In this manuscript, the authors implement a comprehensive isotope-labeling strategy to broadly profile the small-molecule metabolome actively synthesized during blood-stage infection by *Plasmodium falciparum* malaria parasites. Through use of 10 different ¹³C-labelled precursors, multiple extraction conditions, and data filtering against unlabeled erythrocytes and high-abundance metabolites, the authors provide evidence for novel biosynthetic capability by parasites, some of which they explore in more depth via targeted knockdown of four annotated but poorly studied parasite proteins.

Many prior studies have queried parasite biosynthesis of general or specific metabolites but have generally relied on a single ¹³C-labelled precursor (e.g., glucose) or solvent conditions (e.g., MeOH-soluble metabolites). The present study appears to provide the most comprehensive picture of active parasite-dependent metabolite synthesis in blood-stage trophozoites and will serve as an invaluable resource to the parasitology community, especially if the data is organized and presented in a community-accessible manner. Modest insights are made with respect to individual pathways, but these advances seem incremental relative to the broad impact this study will have as a comprehensive metabolite resource. Insights provided by the knock-out/down studies of the tested proteins are intriguing but preliminary and not developed in depth.

Major points:

1. As a comprehensive resource, how will the authors make this data available to the community? The searchable human metabolome database (<https://hmdb.ca>) has provided a valuable resource.

Can the data from the present study be integrated into PlasmoDB.org or elsewhere in an easily searchable format? Such a resource is sorely lacking for the parasitology community.

We have been in discussion with EuPathDB, the curator/host of PlasmoDB and they will host this dataset.

2. The authors limit their study to trophozoite-stage parasites labelled for 5 hours. How were parasites synchronized, and what is the synchrony window? How does this focus on trophozoites and 5-hour labeling potentially bias or limit the observations regarding active pathways and detectable metabolites? Some discussion of this point in the paper would be helpful to justify the authors' choice of these conditions and to contextualize the present observations into the parasite lifecycle.

We have added additional detail regarding parasite synchronization in the methods section. We acknowledge that our focus on trophozoite stages may have led to some minor metabolic pathways/metabolites that are only expressed in other stages of asexual development being overlooked. However, based on our previous metabolomic analyses of different *Plasmodium* developmental stages (Cobbold et al (2013) JBC, McRae et al (2013) BMC Biology, Srivastava et al (2016) PloS Pathogens, Dumont et al (2019) MBio, Kennedy et al (2019) PloS Biology, Yang et al (2019) Cell Reports), we believe that most metabolic pathways are constitutively expressed to some extent throughout asexual development and in many cases are maximally expressed in the rapidly dividing trophozoite stages. The latter stage was also selected for practical reasons; including the fact that infected RBC containing this stage can be rapidly purified, and multi-plex labelling studies can be undertaken over a broad time window allowing isotopic equilibrium in many ¹³C-precursors. While multi-plex labelling of the different asexual RBC stages of *P. falciparum* (with 10 ¹³C-precursors) was beyond the scope of this study, we believe that with the newly developed workflow, this should now be possible. We have included some additional text within the discussion to contextualise our approach.

3. Related to point #2, the impact of this study as a baseline data-set could be improved and clarified by explicit discussion of frontier conditions or questions that can now be dissected via comparative metabolomics. For example, how does de novo metabolite synthesis versus scavenging change in ring or schizont-stage parasites? Given evidence for metabolite scavenging, how does blocking new permeability pathways (e.g., furosemide) or specific knock-down of Exp2, which was recently suggested (<https://pubmed.ncbi.nlm.nih.gov/30150733/>) to mediate metabolite uptake into parasites independent of its role in protein export, alter metabolic profiling? The authors don't need to perform these studies but some discussion of critical applications of the methods herein to address novel questions moving forward would be helpful and place the impact of the present study and advances in context.

We thank reviewer 2 for this very useful feedback. We have updated the manuscript and separated the results and discussion sections to allow further detailed discussion of our findings and how they can be used to further investigate key outstanding questions in the malaria research field.

4. The authors' use of dual loxP/Cre recombinase plus glmS ribozyme for gene regulation is interesting. What efficiency for gene excision did the authors observe? Nothing is reported. For the protein knockdown in figure 4A, was gene excision or transcript degradation more critical for the observed loss of protein levels?

We thank the reviewer for raising this point. This study was undertaken within a larger gene knockdown project, targeting 18 metabolic enzymes for disruption. As such, initial tests were

performed to determine time for depletion and effectiveness of the loxP/Cre and ribozyme system independently. For example, HAD4 appears to be more susceptible to loxP excision than the ribozyme but using both together produces a greater loss of HAD4 in a shorter amount of time.

However, having validated the dual knockdown system, the intention was to utilise the system to induce robust protein depletion independent of which gene/protein was targeted. Hence, the effectiveness of depletion was only assessed at the protein level with both systems activated for the remaining gene targets presented in this study (glucosamine and rapamycin). We have amended the methods to describe this approach.

Minor points:

5. For figure 4H, the 2-4-fold variation in specific activities observed for HAD4 for the profiled monophosphate versus diphosphate substrates seems negligible and much, much smaller than the orders of magnitude difference in substrate specificities reported for optimized phosphate mono- versus di-esterases (e.g., <https://pubs.acs.org/doi/10.1021/bi060847t>). This small difference raises questions about enzyme purity/activity, the kinetic regime (saturating or sub-saturating) in which substrate activities were profiled, and whether the observed specific activities (as opposed to more rigorous determination of specificity constants, k_{cat}/K_m) are an accurate reflection of substrate specificity. The manuscript does not depend critically on this data but some discussion of these limitations and comparison of observed HAD4 activity to well-characterized enzymes in this superfamily would be helpful.

We thank reviewer 2 for raising this point. HAD4 purity was confirmed via Coomassie-stained SDS gels and the band of interest excised and identified via LC-MS. We have now included the SDS gel as an extended figure for completeness. We are confident that the conditions tested are saturating for substrate based on previous work in the Odom-John laboratory, where it was demonstrated that increasing the substrate concentration above 1mM did not increase the hydrolysis rate in the presence of 100nM enzyme. However, we agree with the reviewer that a full kinetic analysis would be more rigorous but beyond the scope of the study.

We have amended the text to highlight the limitations of this approach. In particular, the assay measures free monophosphate release and is not suited to detecting polyphosphate release (and so may not pick up diesterase activity (R-OPO₃-R) of, for example nucleotides that are hydrolyzed with release of polyphosphate products). We have now included in the text details of this limitation. However, it should be noted that the HAD superfamily is structurally and evolutionary distinct from the alkaline phosphatase family and likely to be primarily active against phosphate mono-esters.

We have also added additional comments in the text noting that dissection of the activity of HAD enzymes on nucleotides, such as GDP is complex. Specifically, hydrolysis of GDP by HAD4 would lead to production of GMP, which is also a substrate for the enzyme, resulting in up to double the amount of Pi release, and over estimation of enzyme activity. However, it is worth noting that a similar substrate screen has been performed on other HAD proteins in *E. coli* (10.1074/jbc.M605449200) and yeast (10.1074/jbc.M115.657916), including the nucleotide consuming YrfG from *E. coli* and YrfG's preference for GMP over GDP is of a similar range HAD4's preference for dGMP over dGDP (7 and 6 fold respectively).

6. Lines 315-316: A recent study identified the apicoplast-targeted pyruvate kinase II as likely critical for maintaining NTP pools in the apicoplast (<https://pubmed.ncbi.nlm.nih.gov/32815516/>). The authors may wish to mention and cite this study.

We thank reviewer 2 for pointing this out. We have now included further discussion on the possible role of AMR1 and this recent publication.

Reviewer #3:

The major contribution of this paper is a thorough analysis of the metabolic capacity of Plasmodium-infected red blood cells and providing experimental evidence of a majority of putative metabolic pathways based on gene annotations. This dataset will be an important resource for gene-specific studies of metabolic enzymes, for example as targets of small molecules or identification of new metabolic pathways. Overall this is an important addition to metabolic network analysis of Plasmodium parasites. Several enzymes with unassigned functions are characterized with new hypotheses towards their cellular roles.

Specific comments and questions--

Abstract: The authors state "leading to identification of previously uncharacterized enzymes in isoprenoid biosynthesis, lipid homeostasis, and mitochondrial metabolism." Unfortunately because enzyme depletion can have indirect cellular effects, there is insufficient evidence that the enzyme acts directly in these pathways. Please modify statement as it could be misleading to suggest biochemical activity that has not been proven for these new enzymes.

We thank reviewer 3 for raising this point. We have amended the abstract to the following - 'Validation of the draft metabolome revealed four previously uncharacterised enzymes **which impact** isoprenoid biosynthesis, lipid homeostasis, and mitochondrial metabolism and are necessary for parasite development'

121-125; Fig2. While the identification of ¹³C-labeled metabolites is well explained, it is not clear how unlabeled compounds are verified. As stated in the introduction, "The absence of autonomous methods for controlling the false-discovery rate, has hampered the compilation of an accurate metabolome for most organisms to date." So absent isotope incorporation how are unlabeled metabolites specifically being identified?

Related to this, how many of the metabolites identified were ^{13}C -labeled vs unlabeled?

We have clarified the methods section to further describe this aspect of the study. Briefly, unlabelled metabolites were identified as the final iteration of the study, whereby the 'theoretical' metabolome was used as a reference to search for putative mass-to-charge features that were not previously identified via the stable-isotope labelling experiments. This search operated as a quality control check to ensure that metabolites that should incorporate ^{13}C from a given labelled substrate were not missed (false negatives). Moreover, this metabolite set was useful in confirming genuine 'no label' substrates, such as cofactors, vitamins. Putative metabolite identities were confirmed with pure standards or MS/MS spectral matching.

It should be acknowledged that no publicly available spectral database is complete and there were instances where no MS/MS spectrum was available to match the observed spectrum. Here we manually curated the spectrum to determine if it matches an in silico MS/MS spectrum (METLIN). To aid interpretation we have reported metabolite identification in a tiered structure. The identification list ranks highest confidence (1; which incorporates in ^{13}C -labelling, std matching, and ms2 spectral matching), high confidence (2; std matching or MS/MS spectral matching), medium confidence (^{13}C -labelling) and minimum confidence (4; exact mass matching only).

Of the total 911 metabolites reported for *P. falciparum*-infected erythrocytes, 183 did not label with any ^{13}C -substrate tested and 728 metabolites exhibited ^{13}C -incorporation from at least one ^{13}C -substrate.

187-247: Is it possible these side reaction and damaged products are occurring in dying or lysed cells? Please discuss this possibility.

We thank the reviewer for raising this important issue. We have several lines of evidence to suggest this is unlikely. First, we have established a protocol for purifying trophozoite-stage infected erythrocytes with minimal loss of host cells (<0.1%), and are viable when returned to standard cultivation, proliferating at the standard rate observed with non-purified *P. falciparum* cultures. Secondly, we allow a recovery period (1hr) in standard culturing conditions at very low hematocrit (0.1%) to ensure any metabolic perturbation induced during the 15 minute enrichment process has been overcome and metabolism returned to equilibrium. We have verified the validity of this approach via comparison to unenriched *P. falciparum* cultures, using the ATP/ADP/AMP ratios and the abundance of unique parasite metabolites normalised to parasitaemia. Thirdly, parasites and other eukaryotes harbor enzymes involved in the detoxification of several 'damaged' metabolites (as reviewed 10.1007/s10545-012-9571-1 and 10.1038/nchembio.1141) indicate that they are produced endogenously in intact cell systems. Finally, we have shown that disruption of parasite metabolite repair enzymes leads to marked increases in intracellular pools of these metabolites in viable parasites (10.1128/mBio.02060-19), again indicating that their production occurs in healthy cells, and is normally maintained at low levels by these enzymes.

Note: My expertise is in metabolic pathways and Plasmodium parasites] biology. I am less familiar with technical aspects of metabolomics mass spectrometry. Please refer to other experts for evaluating these aspects of the work.

Thank you again for sending us your revised manuscript. We have now heard back from reviewer #1 who was asked to evaluate your study. As you will see below, the reviewer is satisfied with the modifications made and is supportive of publication.

Before we can formally accept the manuscript for publication we would ask you to address a few remaining editorial issues listed below.

REFEREE REPORTS

Reviewer #1:

all the concerns and requests for clarification have been addressed to the referee's satisfaction

The Authors have now addressed all of the editorial issues.

Accepted

4th Feb 2021

Thank you again for sending us your revised manuscript. We are now satisfied with the modifications made and I am pleased to inform you that your paper has been accepted for publication.

Corresponding Author Name: Malcolm J McConville

Manuscript Number: MSB-20-10023